# A promoter interaction map for cardiovascular disease genetics

**Lindsey E Montefiori[1]\*, Debora R Sobreira[1], Noboru J Sakabe[1], Ivy Aneas[1], Amelia C Joslin[1], Grace T Hansen[1], Grazyna Bozek[1], Ivan P Moskowitz[1,2], Elizabeth M McNally[3], Marcelo A Nóbrega[1]\***

[1]Department of Human Genetics, The University of Chicago, Chicago, United States; [2]Department of Pediatrics and Pathology, The University of Chicago, Chicago, United States; [3]Center for Genetic Medicine, Northwestern University Feinberg School of Medicine, Chicago, United States

**Abstract** Over 500 genetic loci have been associated with risk of cardiovascular diseases (CVDs); however, most loci are located in gene-distal non-coding regions and their target genes are not known. Here, we generated high-resolution promoter capture Hi-C (PCHi-C) maps in human induced pluripotent stem cells (iPSCs) and iPSC-derived cardiomyocytes (CMs) to provide a resource for identifying and prioritizing the functional targets of CVD associations. We validate these maps by demonstrating that promoters preferentially contact distal sequences enriched for tissue-specific transcription factor motifs and are enriched for chromatin marks that correlate with dynamic changes in gene expression. Using the CM PCHi-C map, we linked 1999 CVD-associated SNPs to 347 target genes. Remarkably, more than 90% of SNP-target gene interactions did not involve the nearest gene, while 40% of SNPs interacted with at least two genes, demonstrating the importance of considering long-range chromatin interactions when interpreting functional targets of disease loci.

DOI: https://doi.org/10.7554/eLife.35788.001

**\*For correspondence:**
lem@uchicago.edu (LEM);
nobrega@uchicago.edu (MAN)

**Competing interests:** The authors declare that no competing interests exist.

## Introduction

A major goal in human genetics research is to understand genetic contributions to complex diseases, specifically the molecular mechanisms by which common DNA variants impact disease etiology. Most genome-wide association studies (GWAS) implicate non-coding variants that are far from genes, complicating interpretation of their mode of action and correct identification of the target gene (*Maurano et al., 2012*). Mounting evidence suggests that disease variants disrupt the function of *cis*-acting regulatory elements, such as enhancers, which in turn affects expression of the specific gene or genes that are functional targets of these elements (*Wright et al., 2010*; *Musunuru et al., 2010*; *Cowper-Sal lari et al., 2012*; *Smemo et al., 2014*; *Claussnitzer et al., 2015*). However, because *cis*-acting regulatory elements can be located kilobases (kb) away from their target gene(s), identifying the true functional targets of regulatory elements remains challenging (*Smemo et al., 2014*).

Chromosome conformation capture techniques such as Hi-C (*Lieberman-Aiden et al., 2009*) enable the genome-wide mapping of long-range chromatin contacts and therefore represent a promising strategy to identify distal gene targets of disease-associated genetic variants. Recently, Hi-C maps have been generated in numerous human cell types including embryonic stem cells and early embryonic lineages (*Dixon et al., 2012*, *2015*), immune cells (*Rao et al., 2014*), fibroblasts (*Jin et al., 2013*) and other primary tissue types (*Schmitt et al., 2016*). However, despite the increasing abundance of Hi-C maps, most datasets are of limited resolution (>40 kb) and do not precisely identify the genomic regions in contact with gene promoters.

**eLife digest** Our genomes contain around 20,000 different genes that code for instructions to create proteins and other important molecules. When changes, or mutations, occur within these genes, malfunctioning proteins that are damaging to the cell may be produced. Researchers of human genetics have tried to spot the genetic mutations that are associated with illnesses, for example heart diseases. However, they found that most of these mutations are actually located outside of genes, in the 'non-coding' areas that make up the majority of our genome. These mutations do not modify proteins directly, which makes it challenging to understand how they may be related to heart conditions.

One possibility is that the genetic changes affect regions called enhancers, which control where, when and how much a gene is turned on by physically interacting with it. Mutations in enhancers could lead to a gene producing too much or too little of a protein, which might create problems in the cell. Yet, it is difficult to match an enhancer with the gene or genes it controls. One reason is that a non-coding region can influence a gene placed far away on the DNA strand. Indeed, the long DNA molecule precisely folds in on itself to fit inside its compartment in the cell, which can bring together distant sequences.

Montefiori et al. take over 500 non-coding areas, which can carry mutations associated with heart diseases, and use a technique called Hi-C to try to identify which genes these regions may control. The tool can model the 3D organization of the genome, and it was further modified to capture only the regions of the genome that contain genes, and the DNA sequences that interact with them, in human heart cells.

This helped to create a 3D map of 347 genes which come in contact with the non-coding areas that carry mutations associated with heart diseases. In fact, deleting those genes often causes heart disorders in mice.

In addition, Montefiori et al. reveal that 90% of the non-coding regions examined were influencing genes that were far away. This shows that, despite a common assumption, enhancers often do not regulate the coding sequences they are nearest to on the DNA strand.

Pinpointing the genes regulated by the non-coding regions involved in cardiovascular diseases could lead to new ways of treating or preventing these conditions. The 3D map created by Montefiori et al. may also help to visualize how the genetic information is organized in heart cells. This will contribute to the current effort to understand the role of the 3D structure of the genome, especially in different cell types.

DOI: https://doi.org/10.7554/eLife.35788.002

More recently, promoter capture Hi-C (PCHi-C) was developed which greatly increases the power to detect interactions involving promoter sequences (*Schoenfelder et al., 2015*; *Mifsud et al., 2015*). PCHi-C in different cell types identified thousands of enhancer-promoter contacts and revealed extensive differences in promoter architecture between cell types and throughout differentiation (*Schoenfelder et al., 2015*; *Mifsud et al., 2015*; *Javierre et al., 2016*; *Freire-Pritchett et al., 2017*; *Rubin et al., 2017*; *Siersbæk et al., 2017*). These studies collectively demonstrated that genome architecture reflects cell identity, suggesting that disease-relevant cell types are critical for successful interrogation of the gene regulatory mechanisms of disease loci.

In support of this notion, several recent studies utilized high-resolution promoter interaction maps to identify tissue-specific target genes of GWAS associations. Javierre et al. generated promoter capture Hi-C data in 17 primary human blood cell types and identified 2604 potentially causal genes for immune- and blood-related disorders, including many genes with unannotated roles in those diseases (*Javierre et al., 2016*). Similarly, Mumbach et al. interrogated GWAS SNPs associated with autoimmune diseases using HiChIP where they identified ~10,000 promoter-enhancer interactions that linked several hundred SNPs to target genes, most of which were not the nearest gene (*Mumbach et al., 2017*). Importantly, both studies reported cell-type specificity of SNP-target gene interactions.

Cardiovascular diseases, including cardiac arrhythmia, heart failure, and myocardial infarction, continue to be the leading cause of death world-wide. Over 50 GWAS have been conducted for

these specific cardiovascular phenotypes alone, with more than 500 loci implicated in cardiovascular disease risk (NHGRI GWAS catalog, https://www.ebi.ac.uk/gwas/), most of which map to non-coding genomic regions. To begin to dissect the molecular mechanisms by which genetic variants contribute to CVD risk, a comprehensive gene regulatory map of human cardiac cells is required. Here, we present high-resolution promoter interaction maps of human iPSCs and iPSC-derived cardiomyocytes (CMs). Using PCHi-C, we identified hundreds of thousands of promoter interactions in each cell type. We demonstrate the physiological relevance of these datasets by functionally interrogating the relationship between gene expression and long-range promoter interactions, and demonstrate the utility of long-range chromatin interaction data to resolve the functional targets of disease-associated loci.

## Results

### iPSC-derived cardiomyocytes provide an effective model to study the architecture of CVD genetics

We used iPSC-derived CMs (*Burridge et al., 2014*) as a model to study cardiovascular gene regulation and disease genetics. The CMs generated in this study were 86–94% pure based on cardiac Troponin T protein expression and exhibited spontaneous, uniform beating (*Figure 1—figure supplement 1A*, *Video 1*). To demonstrate that iPSCs and CMs recapitulate transcriptional and epigenetic profiles of matched primary cells, we conducted RNA-seq and ChIP-seq for the active enhancer mark H3K27ac in both cell types and compared these data with similar cell types from the Epigenome Roadmap Project (*Kundaje et al., 2015*). RNA-seq profiles of iPSCs clustered tightly with H1 embryonic stem cells, whereas CMs clustered with both left ventricle (LV) and fetal heart (FH) profiles (*Figure 1—figure supplement 1B*). Furthermore, we observed that matched cell types exhibited three-fold greater overlap in the number of promoter-distal H3K27ac ChIP-seq peaks than non-matched cell types (*Figure 1—figure supplement 1C,D*), indicating that both iPSCs and CMs recapitulate tissue-specific epigenetic states of human stem cells and primary cardiomyocytes, respectively.

To further validate our system, we analyzed differentially expressed genes between iPSCs and CMs. Among the top 10% of over-expressed genes in CMs were genes directly related to cardiac function including essential cardiac transcription factors (*GATA4*, *MEIS1*, *TBX5*, and *TBX20*) and differentiation products (*TNNT2*, *MYH7B*, *MYL7*, *ACTN2*, *NPPA*, *HCN4*, and *RYR2*) (fold-change >1.5, $P_{adj}$ <0.05, *Figure 1—figure supplement 2A–C*). Gene Ontology (GO) enrichment analysis for genes over-expressed in CMs relative to iPSCs further confirmed the cardiac-specific phenotypes of these cells with top terms relating to the development of the cardiac conduction system and cardiac muscle cell contraction (*Figure 1—figure supplement 2D*).

### Promoter-capture Hi-C identifies distal regulatory elements in iPSCs and CMs

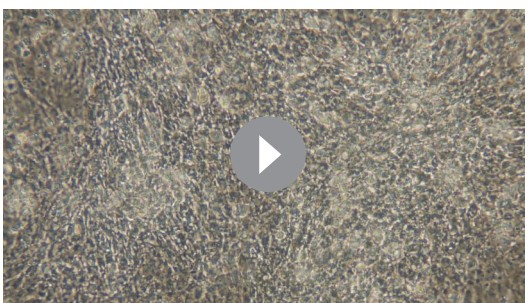

**Video 1.** Video of iPSC-derived cardiomyocytes exhibiting spontaneous beating at day 20 of the differentiation (day of cell harvesting).
DOI: https://doi.org/10.7554/eLife.35788.007

To comprehensively map long-range regulatory elements in iPSCs and CMs, we performed in-situ Hi-C (*Rao et al., 2014*) in triplicate iPSC-CM differentiations; importantly, we used the four-cutter restriction enzyme MboI which generates ligation fragments with an average size of 422 bp, enabling enhancer-level resolution of promoter contacts. We enriched iPSC and CM in situ Hi-C libraries for promoter interactions through hybridization with a set of 77,476 biotinylated RNA probes ('baits') targeting 22,600 human RefSeq protein-coding promoters (see Materials and methods) and sequenced each library to an average depth of ~413 million (M) paired-end reads. After removing duplicates and read-pairs that did not map to a bait, we

obtained an average of 31M and 41M read-pairs per replicate for iPSC and CM, respectively. We used CHiCAGO (*Cairns et al., 2016*), a computational pipeline which accounts for bias from the sequence capture, to identify significant interactions and further filtered for those significant in at least two out of three replicates (see Materials and methods). Finally, we exclusively focused on interactions that were separated by a distance of at least 10 kb. This criterion addresses the high frequency of close-proximity ligation evets in Hi-C data, which are difficult to distinguish as random Brownian contacts or functional chromatin interactions (*Cairns et al., 2016*). In total, we identified 350,062 promoter interactions in iPSCs and 401,098 in CMs. A large proportion (~55%) of interactions were shared between the two cell types, indicating that even at high resolution many long-range interactions are stable across cell types (*Figure 1A*). Approximately 20% of all interactions were between two promoters, demonstrating the high connectivity between genes and supporting the recently suggested role of promoters acting as regulatory inputs for distal genes (*Dao et al., 2017*; *Diao et al., 2017*) (*Figure 1B*). Most interactions were promoter-distal, with a median of ~170 kb between the promoter and the distal-interacting region (*Figure 1C*).

To compare the PCHi-C maps with known features of genome organization, we sequenced our pre-capture Hi-C libraries to an average depth of 665M reads per cell type and identified topologically associating domains (TADs) with TopDom (see Materials and methods). TADs are organizational units of chromosomes defined by <1 megabase (Mb) genomic blocks that exhibit high self-interacting frequencies with a very low interaction frequency across TAD boundaries (*Dixon et al., 2012*; *Nora et al., 2012*). Notably, this organization is thought to constrain the activity of *cis*-regulatory elements to target genes within the same TAD, as disruption of TAD boundaries has been shown to lead to aberrant activation of genes in neighboring TADs (*Nora et al., 2012*; *Lupiáñez et al., 2015*; *Franke et al., 2016*; *Symmons et al., 2016*; *Tsujimura et al., 2015*). We found that the majority of PCHi-C interactions occurred within TADs (73 and 77% in iPSCs and CMs, respectively; *Figure 1D* and *Figure 1—figure supplement 3A*). TAD-crossing interactions ('inter-TAD') contained proportionally more promoter-promoter interactions than intra-TAD interactions, and were more likely to overlap promoter-distal CTCF sites; however, they were similarly enriched for looping to distal H3K27ac sites, a mark of active chromatin (*Figure 1—figure supplement 3B–D*). Inter-TAD interactions had slightly lower CHiCAGO scores, reflecting a lower number of reads supporting these interactions, and spanned greater genomic distances than intra-TAD interactions (*Figure 1—figure supplement 3E,F*). Additionally, promoters with inter-TAD interactions were preferentially located close to TAD boundaries (*Figure 1—figure supplement 3G*) and had higher expression levels compared to promoters with intra-TAD interactions, particularly in CMs (*Figure 1—figure supplement 3H*). These observations are consistent with previous studies which demonstrated that highly expressed genes, specifically housekeeping genes, are enriched at TAD boundaries (*Dixon et al., 2012*).

To illustrate the utility of high-resolution PCHi-C interaction maps, we highlight the *GATA4* locus in *Figure 1D and E*. GATA4 is a master regulator of heart development (*Watt et al., 2004*; *Pikkarainen et al., 2004*) and the *GATA4* gene is located in a TAD structure that is relatively stable between iPSCs and CMs (*Figure 1D*). However, PCHi-C identified increased interaction frequencies between the *GATA4* promoter and several H3K27ac-marked regions, including four in vivo validated heart enhancers from the Vista enhancer browser (*Visel et al., 2007*), specifically in CMs and coincident with strong up-regulation of *GATA4* (*Figure 1—figure supplement 2C*). Although TAD-based analyses help define a gene's *cis*-regulatory landscape, high-resolution promoter interaction data provides the resolution necessary to precisely map enhancer-promoter interactions in the context of cellular differentiation.

To validate the CM interaction map as a resource for cardiovascular disease genetics we next extensively characterized several important aspects of genetic architecture in CMs. We compared CMs with iPSCs in each analysis as a measure of cell-type specificity. These analyses serve as benchmarks that build on established features of genome organization and aid interpretations of the roles that long range interactions play in gene regulation.

## Promoter interactions are enriched for tissue-specific transcription factor motifs

Distal enhancers activate target genes through DNA looping, a mechanism that enables distally bound transcription factors to contact the transcription machinery of target promoters

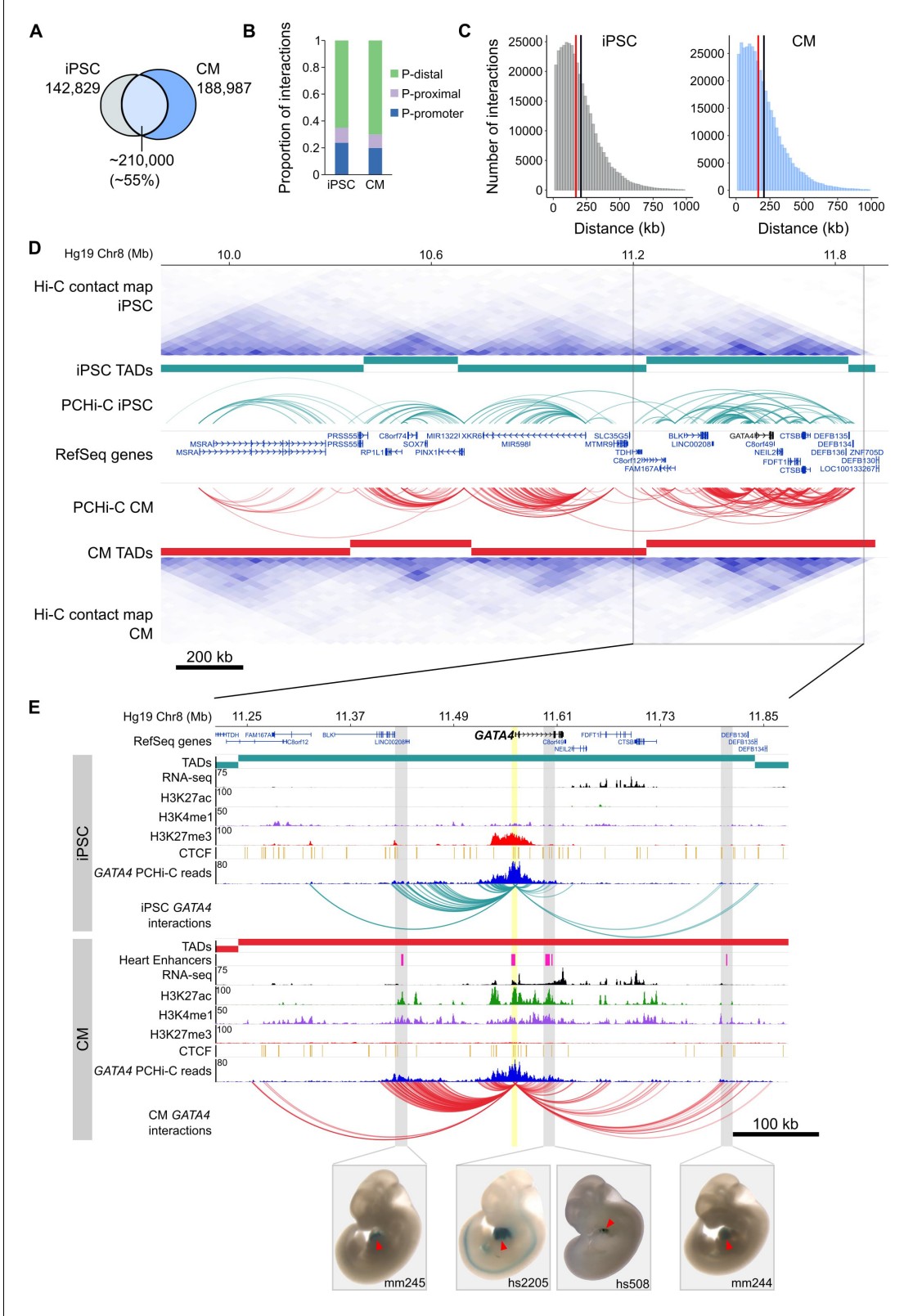

**Figure 1.** General features of promoter interactions. (**A**) Venn diagram displaying the number of cell-type-specific and shared promoter interactions in each cell type. (**B**) Proportion of interactions in each distance category: promoter (P)-promoter (both interacting ends overlap a transcription start site (TSS)); P-proximal (non-promoter end overlaps captured region but not the TSS); P-distal (non-promoter end is outside of captured region). Note that all promoter interactions are separated by at least 10 kb. (**C**) Distribution of the distances spanning each interaction in iPSCs and CMs. The red line

*Figure 1 continued on next page*

Figure 1 continued

depicts the median (170 kb in iPSCs, 164 kb in CMs); the black line depicts the mean (208 kb in iPSCs, 206 kb in CMs). (D) A ~ 2 Mb region of chromosome 8 encompassing the *GATA4* gene is shown along with pre-capture (whole genome) Hi-C interaction maps at 40 kb resolution for iPSCs (top) and CMs (bottom). TADs called with TopDom are shown as colored bars (median TAD size = 640 kb in both cell types, mean TAD size = 742 kb in iPSCs and 743 kb in CMs) and significant PCHi-C interactions as colored arcs. (E) Zoomed-in view of the *GATA4* locus (promoter highlighted in yellow) in iPSCs (top) and CMs (bottom) along with corresponding RNA-seq data generated as part of this study, and ChIP-seq data for H3K27ac, H3K4me1, H3K27me3 and CTCF from the Epigenome Roadmap Project/ENCODE (H1 and left ventricle for iPSC and CM, respectively). Filtered *GATA4* read counts used by CHiCAGO are displayed in blue with the corresponding significant interactions shown as arcs. For clarity, only *GATA4* interactions are shown. Gray highlighted regions show interactions overlapping in vivo validated heart enhancers (pink boxes), with representative E11.5 embryos for each enhancer element (*Visel et al., 2007*). Red arrowhead points to the heart.

DOI: https://doi.org/10.7554/eLife.35788.003

The following figure supplements are available for figure 1:

**Figure supplement 1.** Quality control of iPSC-CMs.
DOI: https://doi.org/10.7554/eLife.35788.004
**Figure supplement 2.** Analysis of RNA-seq in iPSCs and iPSC-CMs.
DOI: https://doi.org/10.7554/eLife.35788.005
**Figure supplement 3.** Analysis of PCHi-C interactions in the context of TADs.
DOI: https://doi.org/10.7554/eLife.35788.006

---

(*Pennacchio et al., 2013*; *Miele and Dekker, 2008*; *Deng et al., 2012*). To assess whether this feature of gene regulation was reflected in the iPSC and CM interactions, we conducted motif analysis using HOMER (*Heinz et al., 2010*) on the set of promoter-distal interacting sequences in each cell type. We initially focused on interactions for genes differentially expressed between iPSCs and CMs (fold-change >1.5, $P_{adj}$ <0.05). We identified CTCF as the most enriched motif in each case (*Figure 2A,B*), consistent with the known role of this factor in mediating long-range genomic interactions (*Phillips and Corces, 2009*; *Phillips-Cremins et al., 2013*; *Nora et al., 2017*). Among the other top motifs, we identified the pluripotency factor motifs OCT4-SOX2-TCF-NANOG (OSN) and SOX2 as preferentially enriched in distal sequences looping to genes over-expressed in iPSCs (*Figure 2A, C*), whereas top motifs in distal sequences looping to genes over-expressed in CMs included TBX20, ESRRB and MEIS (*Figure 2B,C*). TBX20 and MEIS1 transcription factors are important regulators of heart development and function (*Cai et al., 2005*; *Sakabe et al., 2012*; *Mahmoud et al., 2013*) and ESRRB was previously identified as a potential binding partner of TBX20 in adult mouse cardiomyocytes (*Shen et al., 2011*). We also observed that distal interactions unique to either iPSCs or CMs were similarly enriched for tissue-specific transcription factor motifs (*Figure 2D*). In line with a recent report that AP-1 contributes to dynamic loop formation during macrophage development (*Phanstiel et al., 2017*), both iPSC- and CM-specific interactions were enriched for AP-1 motifs (*Figure 2D*), suggesting that AP-1 transcription factors may represent a previously unrecognized genome organizing complex.

## Long-range promoter interactions are enriched for active *cis*-regulatory elements and correspond to gene expression dynamics

Functionally active *cis*-regulatory elements are characterized by the presence of specific histone modifications; active enhancers are generally associated with H3K4me1 and H3K27ac (*Creyghton et al., 2010*; *Heintzman et al., 2009*), whereas inactive (e.g. poised or silenced) elements are often associated with H3K27me3 (*Rada-Iglesias et al., 2011*; *Erceg et al., 2017*). In support of the gene-regulatory function of long-range interactions, we found that the promoter-distal MboI fragments involved in significant promoter interactions were enriched for these three histone modifications in both iPSCs and CMs (*Figure 3A–C*). When promoters were grouped by expression level, we observed that this enrichment increased with increasing expression for H3K27ac and H3K4me1, and decreased with increasing expression for H3K27me3, consistent with an additive nature of enhancer-promoter interactions (*Schoenfelder et al., 2015*; *Javierre et al., 2016*), and validating that PCHi-C enriches for likely functional long-range chromatin contacts.

A strong correlation (Pearson correlation coefficient $r > 0.7$) between the degree of histone modifications and gene expression was first reported nearly 10 years ago (*Karlić et al., 2010*); however, that analysis only considered histone modifications within 2 kb of promoters. To understand whether

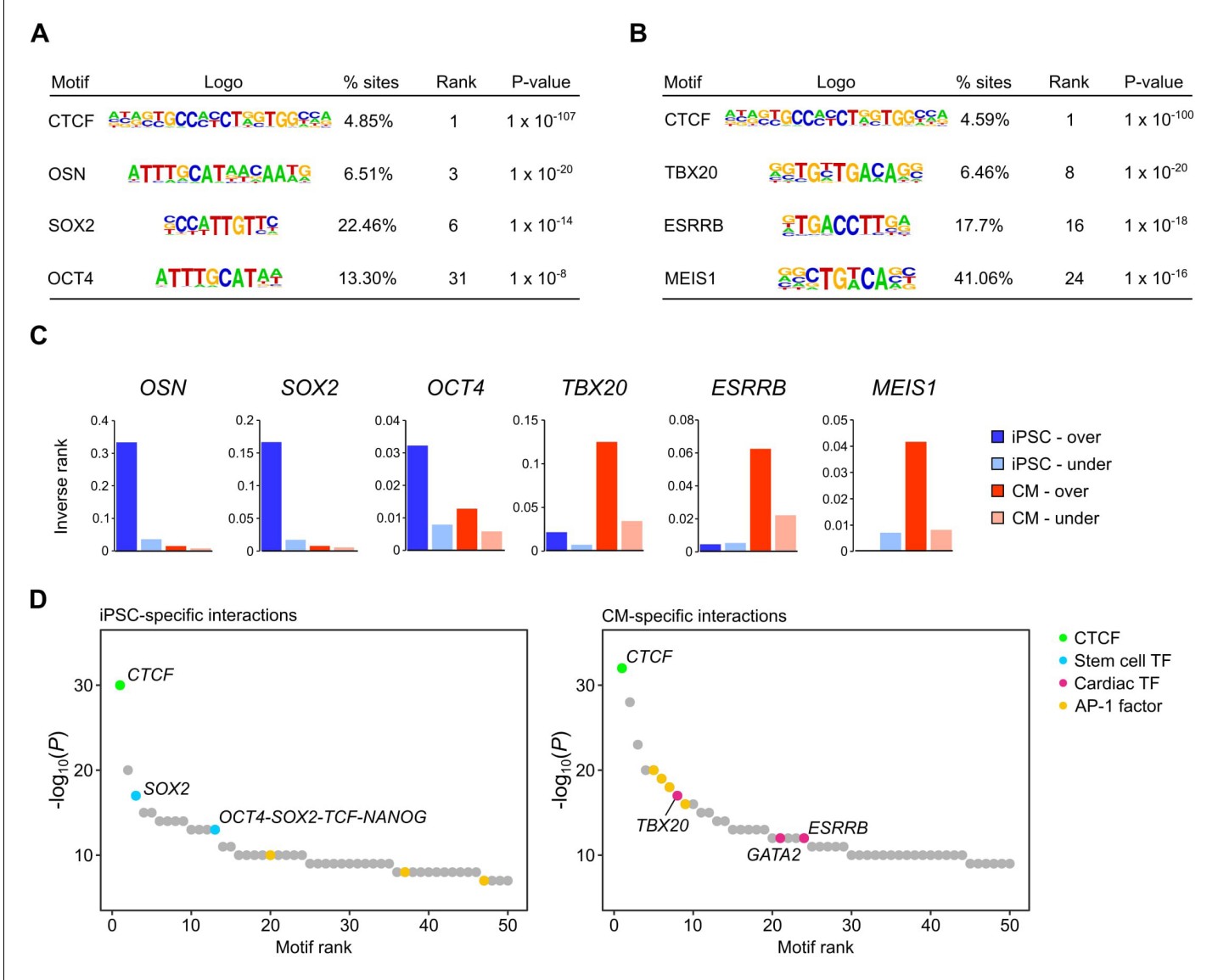

**Figure 2.** Transcription factor motif enrichment in distal interacting regions. (A,B) Selected transcription factor (TF) motifs identified using HOMER in the promoter-distal interacting sequences for all over-expressed genes in (A) iPSCs and (B) CMs (fold change > 1.5, $P_{adj}$ < 0.05). '% sites' refers to the percent of distal interactions overlapping the motif; rank is based on p-value significance. (C) To compare motif ranks across gene sets, the inverse of the rank is plotted for selected motifs identified in distal interactions from over- or under-expressed genes in both iPSCs and CMs. (D) The top 50 motifs identified in cell-type-specific interactions. *OSN*, OCT4-SOX2-TCF-NANOG motif.
DOI: https://doi.org/10.7554/eLife.35788.008

this relationship extends beyond promoter-proximal regions, we correlated the number of histone ChIP-seq peaks within 300 kb of promoters with the promoter's expression level (*Figure 3—figure supplement 1A,B*). H3K27ac and H3K4me1 both positively correlated with expression level (Spearman's $\rho$ = 0.22 and 0.16, respectively in iPSC and $\rho$ = 0.23 and 0.24, respectively in CMs, p<$2.2^{-16}$); in contrast, H3K27me3 negatively correlated with expression level in CMs (Spearman's $\rho$ = −0.20, p<$2.2^{-16}$); however, this relationship was not present in iPSCs (Spearman's $\rho$ = 0.02, p=0.06). Although moderate, these correlations could partially explain why higher expressed genes show stronger enrichment for promoter interactions overlapping histone peaks when using a genome-wide background model (see Materials and methods), and lends support to the notion that active genes are located in generally active genomic environments (*Stevens et al., 2017*; *Gilbert et al., 2004*).

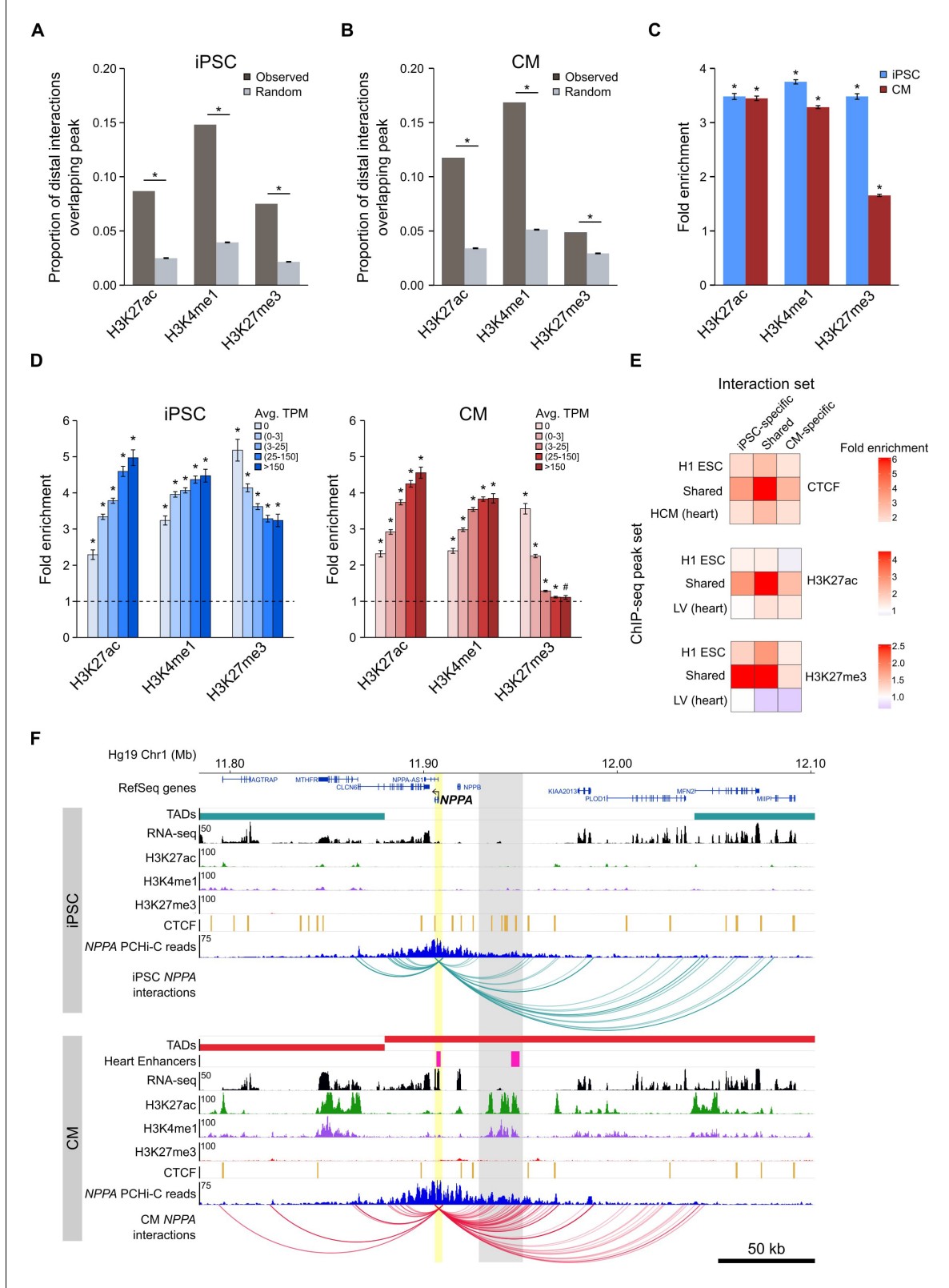

**Figure 3.** Enrichment of promoter interactions to distal regulatory features. (A,B) Proportion of promoter-distal interactions overlapping a histone ChIP-seq peak compared to random control MboI fragments (see Materials and methods). iPSC interactions were overlapped with H1 ESC ChIP-seq data; CM interactions were overlapped with left ventricle ChIP-seq data from the Epigenome Roadmap Project (***Supplementary file 10***). (C) Fold enrichment of the data presented in (A) and (B). (D) Fold enrichment of promoter-distal interactions based on the expression level of the promoter. Promoters were

*Figure 3 continued on next page*

*Figure 3 continued*

grouped into five bins according to their average TPM values. Dashed line indicates no enrichment. (**E**) Fold enrichment of cell-type-specific and shared interactions (columns) to tissue-specific and shared chromatin features (rows). (**F**) Example of the *NPPA* gene in iPSCs (top) and CMs (bottom). Gray box highlights CM-specific interactions to CM-specific chromatin marks and an in vivo heart enhancer (***Visel et al., 2007***). For clarity, only interactions for *NPPA* are shown. *p<0.00001, #p=0.0017, Z-test.

DOI: https://doi.org/10.7554/eLife.35788.009

The following figure supplement is available for figure 3:

**Figure supplement 1.** Correlation between the number of histone ChIP-seq peaks within 300 kb of promoters and gene expression level.

DOI: https://doi.org/10.7554/eLife.35788.010

We next investigated the relationship between cell-type-specific interactions and enrichment for tissue-specific CTCF, H3K27ac, and H3K27me3 marks, hypothesizing that interactions unique to iPSCs or CMs would be most enriched for tissue-specific chromatin features. Indeed, we observed that cell-type-specific interactions preferentially involved H3K27ac peaks from the matched cell type, and were either not enriched (iPSC) or depleted (CM) for H3K27ac marks that were specific to the non-matched cell type (***Figure 3E***, middle panel). However, the strongest enrichment was for cell-type-specific interactions to overlap chromatin features that were present in both cell types (***Figure 3E***). Additionally, interactions that were shared between iPSCs and CMs were most enriched for shared chromatin features. These results suggest that all interactions, whether shared or unique to one cell type, preferentially contact regulatory regions that are active in both cell types, whereas cell-type-specific interactions are not likely to occur in regions specifically marked in the non-matched cell type.

An example of a gene that encompasses these observations is the atrial natriuretic peptide gene *NPPA* (***Figure 3F***) which is specifically expressed in cells of the heart atrium and is upregulated in CMs (***Figure 1—figure supplement 2C***). *NPPA* makes numerous cell-type-specific interactions to a distal region that is only marked with active chromatin (H3K27ac and H3K4me1) in CMs; furthermore, functional characterization showed that this region corresponds to an in vivo enhancer recapitulating *NPPA*'s endogenous expression in the developing heart (***Visel et al., 2007***). Taken together, these results illuminate the complex relationship between long-range promoter interactions and gene regulation and provide evidence that promoter architecture reflects cell-type-specific gene expression.

## Dynamic changes in genomic compartmentalization involve a subset of cardiac-specific genes

As a final benchmark of our datasets, we analyzed large-scale differences in genome organization between iPSCs and CMs. The first Hi-C studies revealed that the genome is organized in two major compartments, A and B, that correspond to open and closed regions of chromosomes, respectively (***Lieberman-Aiden et al., 2009***; ***Rao et al., 2014***). Although most compartments are stable across different cell types, some compartments switch states in a cell-type-specific manner which may reflect important gene regulatory changes (***Dixon et al., 2015***). To assess whether capture Hi-C data, which is more cost-effective for capturing promoter-centered interactions, is able to identify A/B compartments, we compared our capture Hi-C data with pre-capture, genome-wide Hi-C libraries. A/B compartments identified using HOMER (***Heinz et al., 2010***) were remarkably similar in the whole-genome and PCHi-C datasets (97% correspondence, ***Figure 4A***, top panel, and ***Figure 4—figure supplements 1*** and ***2***), demonstrating that PCHi-C data contains sufficient information to identify broadly active and inactive regions of the genome. As an example, we highlight a 10 Mb region on chromosome 4 containing the *CAMK2D* gene locus (***Figure 4A***). Compartments were relatively stable across this region in iPSCs and CMs; however, the *CAMK2D* gene itself was located in a dynamic compartment that switched from inactive in iPSCs to active in CMs. Correspondingly, this gene was highly upregulated during differentiation to CMs (***Figure 4A***, inset).

We observed this effect on a global level, as genes located in A compartments were expressed at significantly higher levels than genes located in the B compartments in both iPSCs and CMs (***Figure 4B***). Additionally, genes that switched A/B compartments between cell types were correspondingly up- or down-regulated (***Figure 4C***). GO analysis of the 1008 genes that switched from B to A compartments during iPSC-CM differentiation revealed enrichment for terms such as

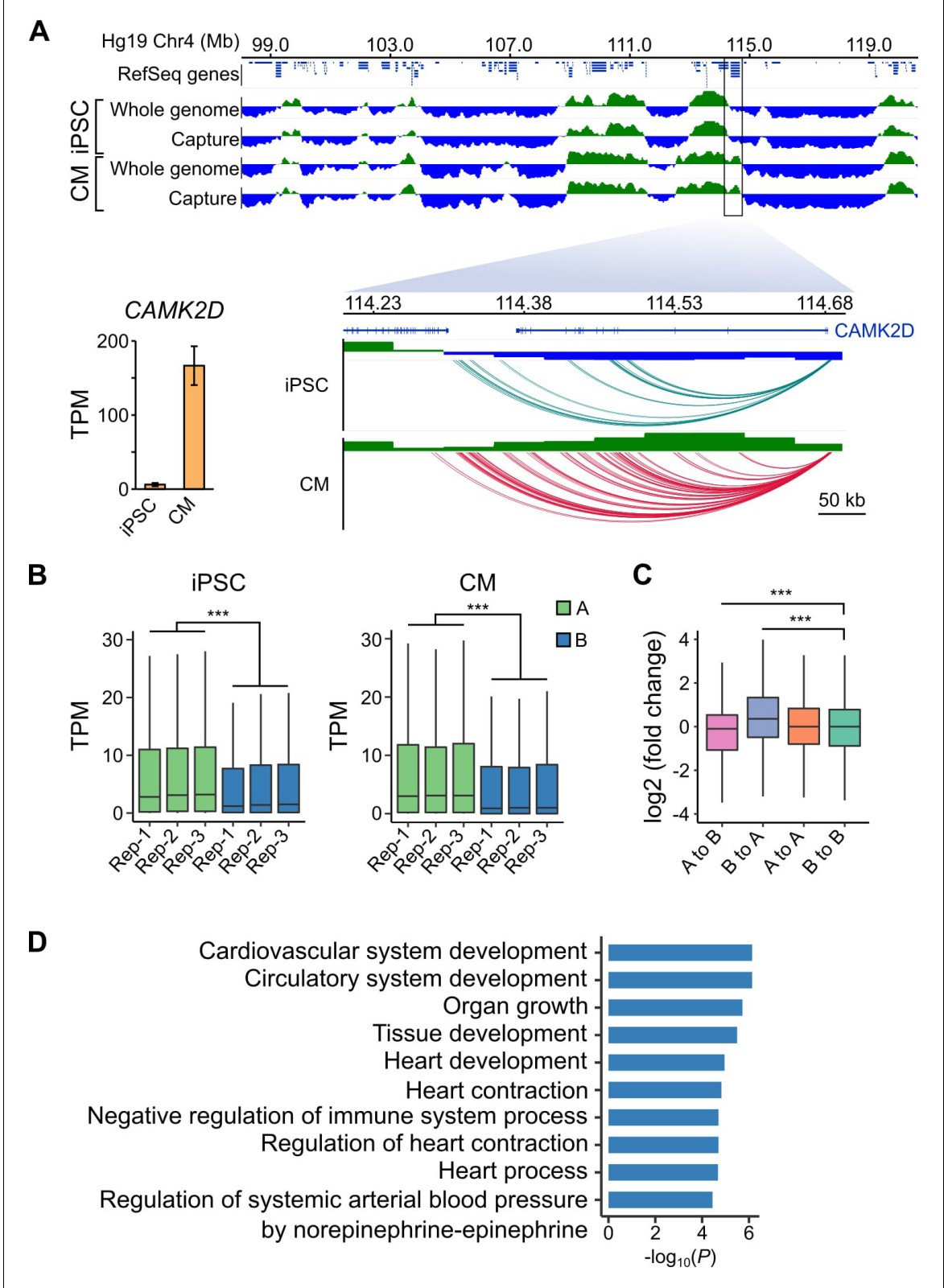

**Figure 4.** A/B compartment switching corresponds to activation of tissue-specific genes. (**A**) Top panel: 10 Mb region on chromosome four showing A (green) and B (blue) compartments based on the first principle component analysis calculated by HOMER (*Heinz et al., 2010*) of the whole-genome Hi-C and capture Hi-C interaction data. Bottom panel: zoomed in on the *CAMK2D* locus; only capture Hi-C A/B compartments shown. Inset: expression level of *CAMK2D* in iPSCs and CMs across the three replicates. (**B**) Expression level (TPM) of genes located in the A (green) or B (blue) compartment in

*Figure 4 continued on next page*

*Figure 4 continued*

each replicate of iPSC (left) or CM (right). (C) Difference in expression level (log2 fold change relative to iPSCs) of genes switching compartments from iPSC to CM or remaining in stable compartments. (D) Gene Ontology analysis of biological processes associated with genes switching from B to A compartments during iPSC-CM differentiation. ***p<2.2 × 10$^{-16}$, Wilcoxon rank-sum test.

DOI: https://doi.org/10.7554/eLife.35788.011

The following figure supplements are available for figure 4:

**Figure supplement 1.** Comparison of A/B compartments in Hi-C and PCHi-C.

DOI: https://doi.org/10.7554/eLife.35788.012

**Figure supplement 2.** Example of A/B compartments.

DOI: https://doi.org/10.7554/eLife.35788.013

**Figure supplement 3.** GO analysis on the genes switching from active A compartments in iPSCs to inactive B compartments in CMs.

DOI: https://doi.org/10.7554/eLife.35788.014

'cardiovascular system development' and 'heart contraction' (*Figure 4D*, *Supplementary file 5*). Importantly, these genes were identified based solely on their location in a dynamic genomic compartment and not from gene expression data. GO analysis for genes that switched from A to B compartments during iPSC-CM differentiation related to non-cardiac processes, such as skin development, epithelial cell differentiation and sex determination (*Figure 4—figure supplement 3*, *Supplementary file 5* and *6*). These data show that PCHi-C accurately captured tissue-specific interactions and indicate that compartmentalization of genes in spatially regulated regions of the nucleus may be one mechanism to ensure tissue-specific gene expression (*Dixon et al., 2015*). In summary, our analyses demonstrated that CM promoter interactions recapitulate key features of cardiac gene regulation and function, validating the CM map as an important tool to investigate CVD genetics.

## CM promoter interactions link GWAS SNPs to target genes

A particularly relevant application of high-resolution promoter interaction maps is to guide post-GWAS studies by identifying the target genes of disease-associated variants. We employed this approach to link GWAS SNPs for several major cardiovascular diseases to their target gene(s) using the CM interaction map. We compiled 524 lead SNPs from the NHGRI database (https://www.ebi.ac.uk/gwas/) for three important classes of CVDs: cardiac arrhythmias, heart failure, and myocardial infarction (*Table 1*, *Supplementary files 7* and *8*). Because of linkage disequilibrium (LD) patterns, the true causal SNP could be any SNP in high LD with the lead variant. Therefore, we expanded this set of SNPs to include all variants in high LD ($r^2$ >0.9, within 50 kb of lead SNP), increasing the number of putatively causal variants to 10,475 (hereafter called LD SNPs). We found that 1999 (19%) of the LD SNPs were located in promoter-distal MboI fragments that interacted with the promoters of

**Table 1.** Summary of the SNPs and target genes characterized in each disease class.
Summary values for each disease group are depicted along with the total number of GWAS, SNPs, and target genes ('Combined' column). Tag SNPs were identified from the published GWAS in the NHGRI-EBI database; SNPs in LD are the total number of non-promoter SNPs (including tag SNPs) in LD ($r^2$ > 0.9) with the tag SNPs in each disease group; SNPs looping to genes are the SNPs in LD that are in a distal promoter interaction; Target genes are all genes with an interaction to a promoter-distal SNP. See *Supplementary file 8* for a complete list of all GWAS, coordinates of each SNP and its assigned target gene, expression level in iPSC and CM, and mouse knock-out phenotype where available.

|  | Arrhythmia | Myocardial infarction | Heart failure | Combined |
|---|---|---|---|---|
| Number of studies | 30 | 11 | 11 | 50 |
| Tag SNPs | 358 | 86 | 80 | 524 |
| SNPs in LD | 6555 | 1822 | 2098 | 10,475 |
| SNPs looping to genes | 1152 | 357 | 490 | 1999 |
| Target genes | 237 | 72 | 53 | 347 |

DOI: https://doi.org/10.7554/eLife.35788.016

347 genes in CMs (*Supplementary file 8*), hereafter referred to as target genes. The majority (89%) of LD SNP-target gene pairs were located within the same TAD, with a median distance of 185 kb between each SNP-target gene pair (*Figure 5A*). Importantly, 90.4% of SNP-target gene interactions skipped at least one gene promoter and 42% of SNPs interacted with at least two different promoters (*Figure 5B*).

To confirm that the CM PCHi-C interactions linked SNPs to CVD-relevant target genes, we performed GO analysis and found that target genes were highly and specifically enriched for biological processes related to cardiac function, such as membrane repolarization and cardiac conduction (*Figure 5C*, left panel and *Supplementary file 5* and *6*). As a control, we used iPSC interactions to link the same SNPs to target genes and observed a completely different set of unrelated biological processes for these genes (*Figure 5C*, right panel). To further characterize the biological relevance of target genes, we mined mouse knock-out data from the Mouse Genome Informatics (MGI) database (*Blake et al., 2017*), which revealed that a statistically significant number of target genes resulted in a cardiovascular phenotype when knocked-out in the mouse (78 genes (22.4%), p=1 × $10^{-5}$, *Figure 5D*). Finally, we examined expression quantitative trait loci (eQTL) data from human left ventricle (LV) tissue obtained as part of the Genotype-Tissue Expresion (GTEx) Project (*Carithers et al., 2015*) and found that of the 1999 LD SNPs in interactions, 410 (20.5%) corresponded to LV eQTLs; in comparison, only 12.2% of the full set of LD SNPs corresponded to LV eQTLs (p<0.00001, *Figure 5E*). We next assessed whether eQTLs loop to their associated gene. For this analysis, we considered the full set of LV eQTLs, as the 410 LD SNP eQTLs represent too small of a proportion of the full set (<0.1% of all LV eQTLs) to fully ascertain significance. On a genome-wide level, LV eQTLs in promoter-distal interactions were significantly more likely to loop to their associated gene than expected by chance (p<0.00001, *Figure 5F*, left panel). Importantly, this significance decreased when LV eQTLs were analyzed with iPSC promoter interactions (p=0.035, *Figure 5F*, right panel). Taken together, these results indicate that CM promoter interactions identify a subset of disease-relevant SNPs most likely to be functional and support the use of the CM map to assign distal CVD-associated SNPs to putative target genes.

## Using gene expression as a metric for interpreting disease-relevance of newly identified target genes

Based on an enrichment of target genes with known cardiac function, we next assessed whether expression level is an informative metric to further prioritize functional follow-up studies. We examined the expression level of the 347 target genes and found that they were moderately over-expressed in CMs compared to iPSCs (median log2 fold change = 1.08, mean log2 fold change = 1.44, mean TPM values were 40.6 in iPSCs and 60.1 in CMs, p=0.12, *Figure 6A and B*). Although not significant, this result reflects the enrichment of known cardiac-related genes that interact with CVD loci. However, because a subset of target genes was over-expressed in iPSCs relative to CMs (*Figure 6C*), we predicted that gene expression level alone may be an insufficient metric to gauge the relevance of target genes to CVD biology. Indeed, we found that 21 of the 78 target genes (27%) that cause cardiovascular phenotypes when knocked-out in mice were overexpressed in iPSCs compared to CMs (*Supplementary file 8*). This result indicates that putatively causal genes may not appear as obvious candidates based solely on gene expression data.

To illustrate this point, we highlight two genes: *TBX5*, a gene directly linked to cardiac arrhythmia (*Figure 6D*) (*Smemo et al., 2012*; *Arnolds et al., 2012*), and *LITAF*, a gene that, until recently, had no obvious role in cardiac biology (*Moshal et al., 2017*) (*Figure 6E*). Both genes formed long-range interactions to LD SNPs identified in arrhythmia GWAS, making both genes candidate functional targets of the GWAS associations. *TBX5*, which is over-expressed in CMs (*Figure 6C*), is the most likely target gene of the LD SNPs nearby based on the interaction data but also because of its known role in directing proper development of the cardiac conduction system. *LITAF*, on the other hand, was over-expressed in iPSCs compared to CMs (*Figure 6C*) and was not known to contribute to cardiac function until a recent study identified this gene as a regulator of cardiac excitation in zebrafish hearts (*Moshal et al., 2017*).

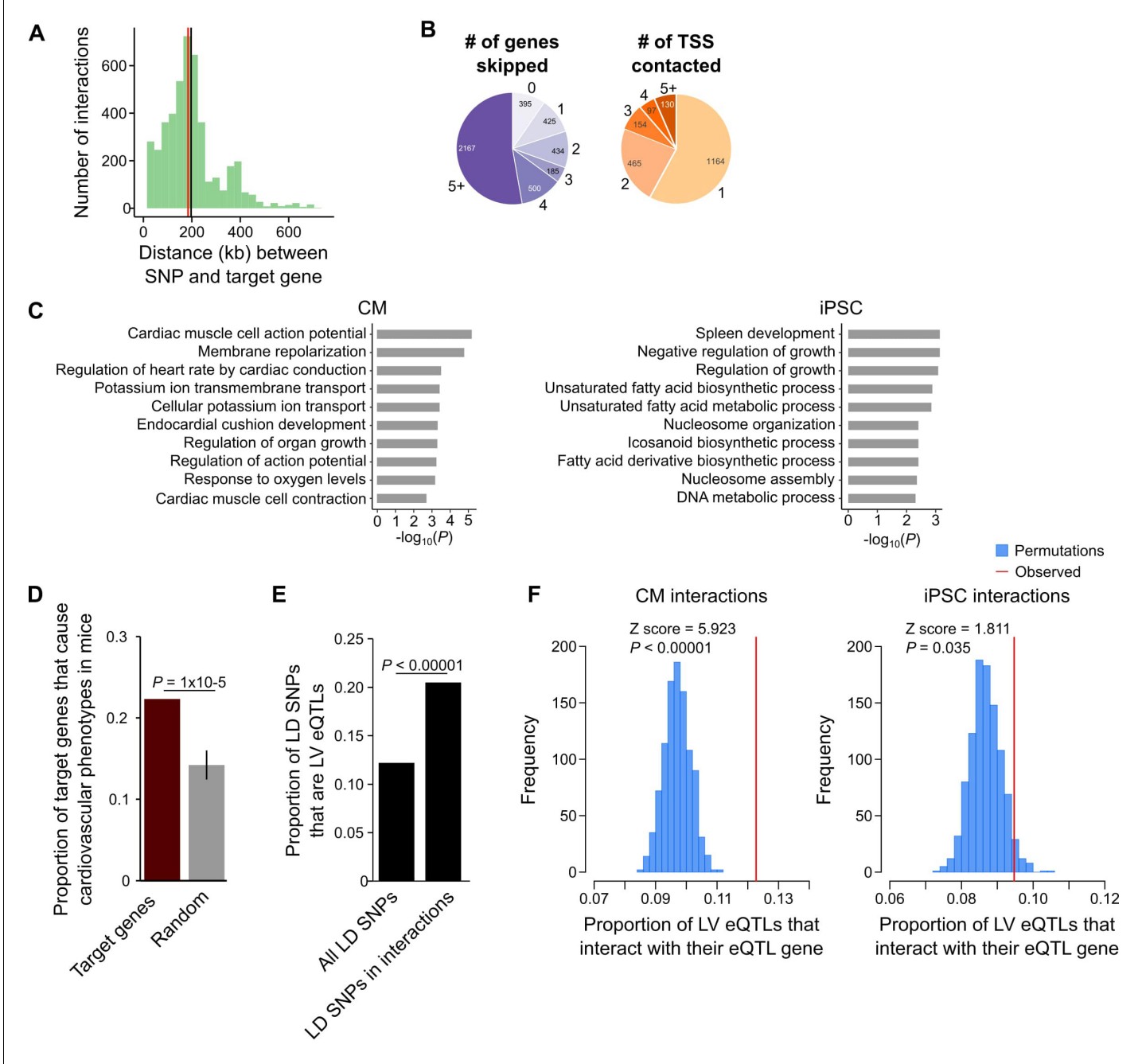

**Figure 5.** CM promoter interactions link CVD GWAS SNPs to target genes. (**A**) Distribution of genomic distances separating SNP-target gene interactions (red line, median = 185 kb; black line, mean = 197 kb). (**B**) Pie chart showing the number of TSS's skipped for each SNP-target gene interaction (left) and the number of genes contacted by each SNP (right). (**C**) GO enrichment analysis for genes looping to LD SNPs using the CM promoter interaction data (left panel) or the iPSC promoter interaction data (right panel). (**D**) Proportion of target genes that result in a cardiovascular phenotype when knocked-out in the mouse (MGI database [**Blake et al., 2017**]), compared to a random control set. p-Value calculated with a Z-test. (**E**) Proportion of GWAS LD SNPs that are eQTLs in left ventricle (LV) when considering either the full set of LD SNPs, or the subset that overlap CM promoter interactions. p-Value calculated with Fisher's exact test. (**F**) Proportion of LV eQTLs (genome-wide) that map within a promoter interaction for the eQTL-associated gene (indicated by the red line). Random permutations were obtained by re-assigning each promoter's set of interactions to a new promoter and calculating the proportion of eQTLs in random interactions that interact with their eQTL-associated gene. Proportions only consider eQTLs that overlap a promoter-distal interaction. P-values calculated with a Z-test.

DOI: https://doi.org/10.7554/eLife.35788.015

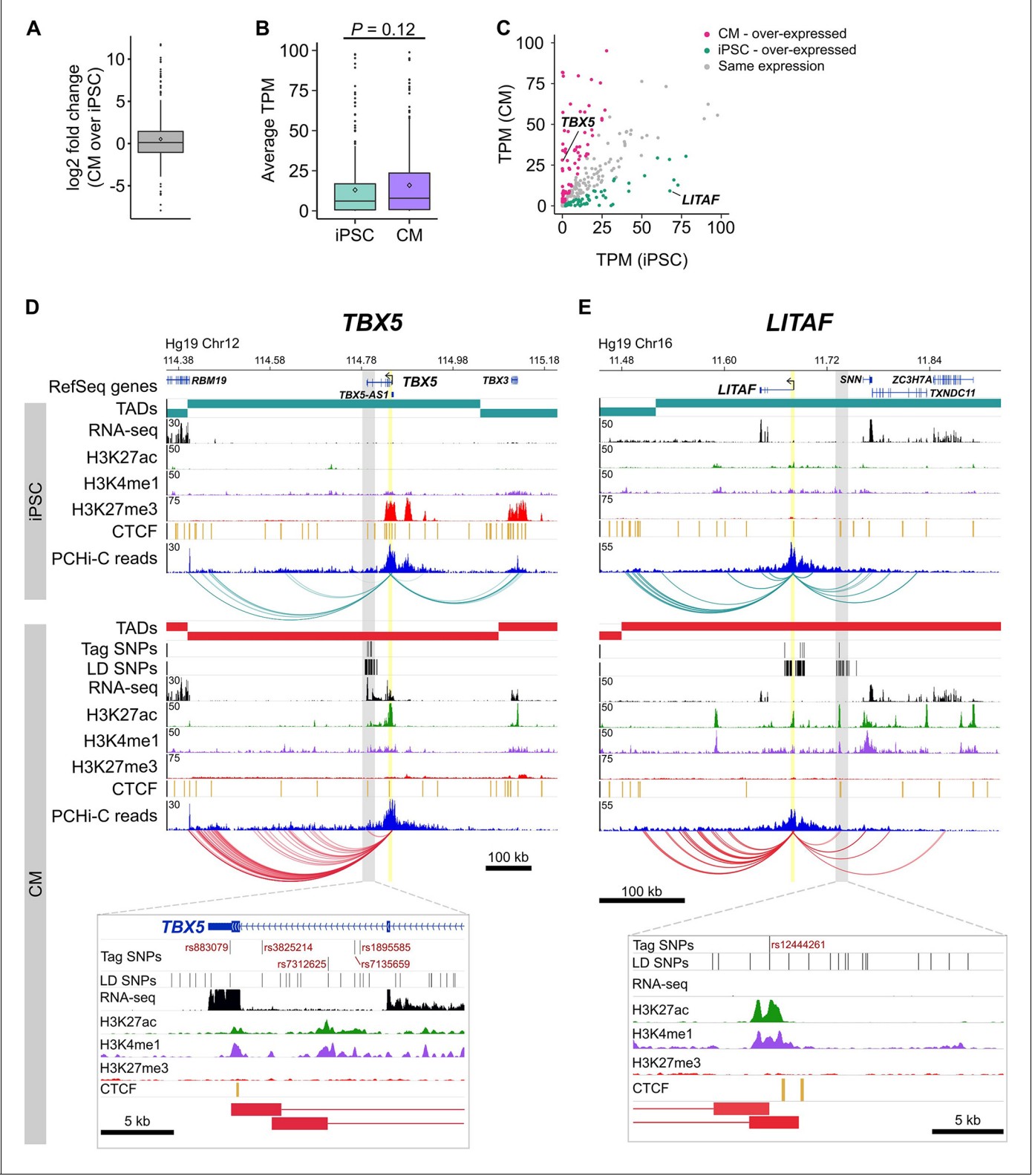

**Figure 6.** Characterizing target genes based on expression level. (**A**) Log2 fold change of the expression level of target genes in CMs compared to iPSCs (horizontal bar indicates median, 1.08; diamond indicates mean, 1.44). (**B**) Average TPM values of target genes in iPSCs and CMs (p=0.12, Wilcoxon rank-sum test). Diamonds indicate the mean value (40.6 for iPSC, 60.1 for CM). (**C**) Comparison of average TPM values for target genes in CMs and iPSCs. See *Supplementary file 8* for full list of genes and TPM values. (**D,E**) Examples of genes looping to cardiac arrhythmia GWAS SNPs in *Figure 6 continued on next page*

*Figure 6 continued*

CMs. (**D**) The *TBX5* gene interacts with a functionally validated arrhythmia locus (*Smemo et al., 2012*). (**E**) The *LITAF* gene interacts with a locus identified in (*Arking et al., 2014*). Yellow highlighted region indicates the promoter; gray box and zoom panel show the promoter-interacting regions (pink boxes) overlapping arrhythmia SNPs. For clarity, only interactions for the indicated promoter are shown.
DOI: https://doi.org/10.7554/eLife.35788.017

## CM promoter interactions are informative to cardiovascular associations that do not directly involve cardiomyocytes

Because the three disease classes that we analyzed represent diverse pathologies, we predicted that the target genes identified for each class individually may relate to different biological processes. Specifically, we considered that cardiac arrhythmias – which directly result from defects in cardiomyocytes specialized for electrical conduction – may uncover the most cardiac-relevant target genes compared to heart failure and myocardial infarction, two CVDs that also involve non-cardiac systems. When broken down into the respective disease classes, we confirmed that the majority of the GO enrichment for cardiac terms was driven by the cardiac arrhythmia SNPs (*Figure 7A*), with terms directly related to the cardiac conduction system. Myocardial infarction (*Figure 7B*) and heart failure (*Figure 7C*) analyses uncovered a set of genes that were slightly enriched for regulation of growth and morphogenesis, respectively.

Despite these seemingly non-specific processes, each set of target genes contained important disease-relevant candidates. For example, one of the strongest associations for myocardial infarction lies in-between the *CELSR2* and *PSRC1* genes on chromosome 1p13, but a careful screen of genes whose expression was affected by the risk allele implicated the more distal *SORT1* gene (*Musunuru et al., 2010*). *SORT1* encodes a sorting receptor that is expressed in many tissues and has been shown to act in the liver to regulate cholesterol levels (*Petersen et al., 1997*; *Musunuru et al., 2010*). Despite functioning in the liver, we identified multiple promoter interactions between *SORT1* and the myocardial infarction GWAS locus in CMs (*Figure 7D*), directly implicating *SORT1* as the target gene and lending further support to experimental validation of this locus as a *SORT1* enhancer (*Musunuru et al., 2010*). Additionally, the *ACTA2* gene is located 220 kb away from the heart failure GWAS locus proximal to the *CH25H* and *LIPA* genes on chromosome 10q21 (*Smith et al., 2010*) (*Figure 7E*). *ACTA2* encodes the smooth muscle cell-specific actin protein and mutations in this gene have been shown to cause coronary artery disease, among other vascular diseases (*Guo et al., 2009*). Despite its location at a considerable distance from the GWAS association, chromatin interactions provide an important level of evidence that *ACTA2* is a putative causal gene in the development of heart failure. Therefore, the CM interaction map is not only useful to interrogate diseases directly related to cardiomyocytes, as in the case of cardiac arrhythmias, but also aids interpretation of target genes that may act in non-cardiac tissues.

## Discussion

Incomplete understanding of long-range gene regulation is a major roadblock in the translation of GWAS-associated loci to disease biology. Major challenges in this process include identifying putatively causal variants mapping within regulatory elements and functionally connecting these regulatory elements to their target genes. To delineate gene-regulatory interactions between CVD-associated SNPs and putative causal genes, we generated high-resolution maps of promoter interactions in human iPSCs and iPSC-derived CMs. We demonstrated that promoters interact with a diverse set of distal DNA elements in both cell types, including known enhancer sequences, which reflect cell identity and correspond to tissue-specific gene expression. To demonstrate the utility of the CM map, we linked 1,999 CVD-associated SNPs to putative causal target genes which identified both validated and potentially novel genes important for cardiovascular disease biology. To validate the biological relevance of our maps, we addressed several important features of long-range chromatin interactions in comparative analyses.

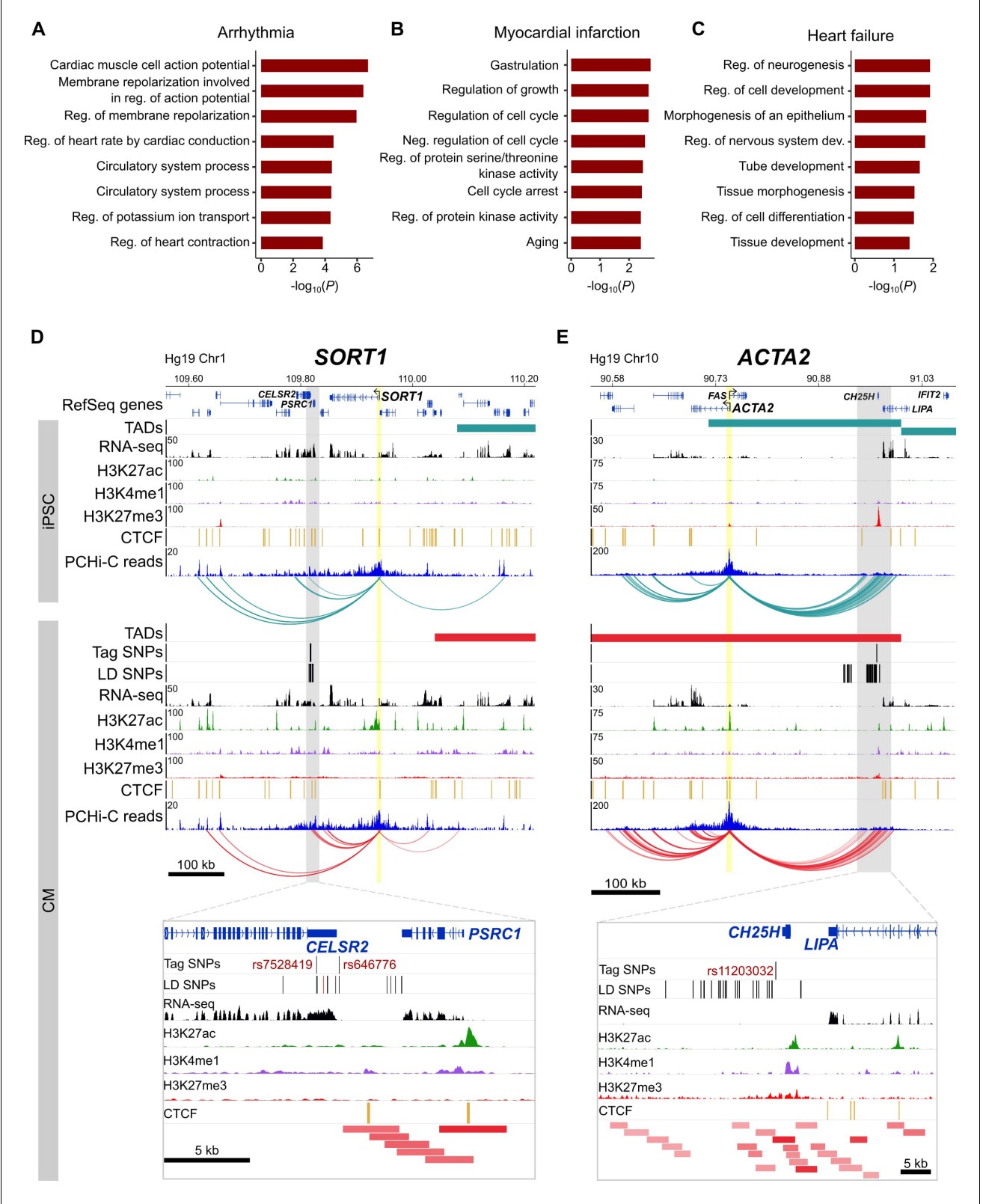

**Figure 7.** Relevance of CM promoter interactions for cardiac arrhythmia, myocardial infarction and heart failure. (**A–C**) Gene Ontology analysis for target genes looping to (**A**) cardiac arrhythmia SNPs, (**B**) myocardial infarction SNPs, and (**C**) heart failure SNPs. (**D**) The *SORT1* promoter loops to a distal myocardial infarction locus (***Musunuru et al., 2010***). The rs12740374 SNP shown to disrupt a C/EBP binding site in (***Musunuru et al., 2010***) is

*Figure 7 continued on next page*

*Figure 7 continued*
colored red. (E) The *ACTA2* promoter loops to the 10q21 heart failure locus (*Smith et al., 2010*). Zoom plots depict the full interacting region overlapping GWAS LD SNPs. For clarity, only interactions for the indicated gene are shown.
DOI: https://doi.org/10.7554/eLife.35788.018

## Promoters contact distal regions enriched for tissue-specific transcription factor motifs

Gene regulation by distant regulatory elements involves the bridging of linearly separated DNA sequences, for example between a promoter and its distal enhancers, through chromatin looping mechanisms (*Spitz and Furlong, 2012*). In support of this model, we report an enrichment of tissue-defining transcription factor motifs in the distally interacting sequences of differentially expressed promoters both for CMs and iPSCs, providing an important level of evidence to validate the functional relevance of iPSC and CM interactions. One explanation for this enrichment is that our interaction maps are high resolution. We generated Hi-C libraries with the 4 bp cutter MboI, which generates fragments with an average size of 422 bp; this increased specificity of the captured region likely leads to better resolution of the underlying enhancer sequence and, consequently, increased power to detect short transcription factor binding motifs.

## Influence of active and repressive promoter interactions on gene expression level

The majority of capture Hi-C studies to date have reported that gene expression level correlates with enrichment for various histone marks. We observed the same trend in our data, with highly expressed genes exhibiting strong enrichment for looping to distal H3K4me1 and H3K27ac-marked regions, and lowly expressed genes exhibiting strong enrichment for looping to H3K27me3-marked regions. These data are consistent with a model in which the number of long-range interactions to enhancers or repressors additively contributes to gene expression level (*Schoenfelder et al., 2015*; *Javierre et al., 2016*). The forces that drive increased association between promoters and distal *cis*-regulatory elements are not completely understood and have been topics of investigation in the genome organization and chromatin biology fields for several years (*Dekker and Mirny, 2016*; *Calo and Wysocka, 2013*). One possibility is that this increasing enrichment is driven by genomic compartmentalization of active and inactive chromatin. We showed that a gene's expression level correlates with the number of histone ChIP-seq peaks within a large window (300 kb) surrounding each promoter. Thus, highly expressed genes are more likely to contact active chromatin regions compared to lowly expressed genes, corresponding to the observed increasing enrichment of contacts and expression we and others have reported. This local increase in active or repressive chromatin may be one driving force underlying the expression level-dependent increase in association between promoters and *cis*-regulatory elements, akin to a phase separation-mediated model of enhancer-promoter interactions (*Hnisz et al., 2017*).

## A promoter interaction map for cardiovascular disease genetics

We demonstrated several ways in which promoter interaction data can be used to better understand disease genetics, specifically addressing the major requirement for a high-resolution map of the gene-regulatory network in human cardiomyocytes. Although iPSC-derived CMs are known to be relatively immature and do not fully reflect the diverse structural and functional aspects of adult cardiac cells (*Gherghiceanu et al., 2011*; *Karakikes et al., 2015*), the difficulty in obtaining pure subpopulations of primary cardiomyocytes with high integrity necessitates the use of an in vitro system. We showed that the CMs used in this study were highly pure and recapitulate known gene regulatory properties of primary cardiomyocytes. Because of this purity, we were able to integrate CVD-associated SNPs with CM promoter interactions with high confidence, assigning nearly 20% of the variants in high LD with these associations to 347 target genes.

Supporting the physiological relevance of CMs to the cardiac conduction system, we found that target genes were most relevant for GWAS loci associated with cardiac arrhythmias, in line with previous findings in immune cells that many target gene interactions were unique to relevant immune cell subtypes (*Javierre et al., 2016*; *Mumbach et al., 2017*). Our data also revealed that even for

diseases whose etiology involves cell types other than cardiomyocytes, such as myocardial infarction and heart failure, we identified interactions involving loci associated with these diseases that recapitulate the enhancer-promoter interactions in non-cardiac cell types. As an example, we showed that a validated myocardial infarction locus interacts with the distal *SORT1* promoter in CMs even though this locus has been extensively characterized in the context of cholesterol metabolism in hepatocytes. Therefore, the promoter interactions we observe linking the disease locus to *SORT1* may represent tissue-invariant genome architecture, likely reflecting that genome organization in general is relatively stable (*Dixon et al., 2015*; *Jin et al., 2013*; *Ghavi-Helm et al., 2014*). While we advocate the use of the CM map for investigating gene regulatory mechanisms of diseases related to cardiomyocyte biology, we also emphasize that, where identified, any interaction between a promoter and a putative disease-associated genomic region serves as an important level of evidence to prioritize that gene for future follow-up studies.

## Limitations of the PCHi-C maps

The PCHi-C technique holds great promise to identify with high resolution and throughput all gene regulatory elements in any tissue or developmental stage of interest. However, due to technical and biological limitations, there are important caveats to PCHi-C that should be considered when interpreting the iPSC or CM interaction data. The most important caveat is that there are likely to be many false negatives, or 'missing' interactions. Although the capture step greatly enriches for promoter-containing ligation fragments in a Hi-C library, the total landscape of promoter contacts in a population of cells is still under-sampled, even with a sequencing depth of ~400M reads per replicate conducted for this study. This is due to several factors, including the hybridization efficiency of each bait, ability to design sufficient baits per promoter, and the transient nature of many regulatory interactions. This latter issue is confounded by the distance-dependent effect on ligation frequency: as the distance between two fragments increases, the read-depth required to robustly identify that interaction also increases. The feasibility of deeper sequencing and modifications to computational pipelines will continue to improve the coverage and resolution of Hi-C data.

Additionally, because the CHiCAGO program does not incorporate TAD boundaries into its background model, it may slightly underestimate the expected number of reads corresponding to intra-TAD interactions which could lead to potential false positives. However, we note that there is a strong correspondence between TADs called on pre-capture Hi-C data and PCHi-C interactions identified with CHiCAGO (*Figure 1—figure supplement 3A*); this suggests that accounting for TAD boundaries may only marginally improve our ability to identify significant interactions.

A final consideration is the interpretation of interactions involving inactive genes. Although most regulatory elements are thought of as activating, it is possible that long-range interactions may also contribute to gene silencing; this is supported by the observation that silent genes are enriched for long-range interactions to H3K27me3 marked regions (*Figure 3D*). Alternatively, silent genes may contact regulatory elements that are not active in the analyzed cell type or developmental stage; these may represent 'pre-formed' loops between genes and their regulatory elements as characterized in *Ghavi-Helm et al. (2014)*.

Despite these limitations, the data sets we provide here represent a highly enriched set of ~350,000 and~400,000 promoter interactions in iPSC and CMs, respectively; although there are likely missing interactions, the interactions that we did identify should be considered as very high confidence, as they were independently identified in at least two biological replicates and show strong signal of enrichment for known features of genome architecture and gene regulation. In conclusion, the promoter interaction maps we generated in this study represent important resources for any investigation into the gene regulatory mechanisms underlying cardiovascular disease traits. The list of candidate regulatory variants and their target genes may serve as an entry point for several hypotheses related to CVD GWAS, and can be readily tested in experimental settings. To provide both the iPSC and CM maps as an accessible resource, we have hosted the full set of data presented in this study as a public track hub at the WashU EpiGenome Browser (*Zhou et al., 2015*), accessible at the following link: http://epigenomegateway.wustl.edu/browser/?genome=hg19&publichub=Lindsey. Additionally, we provide the significant PCHi-C interaction files used in all analyses in the Supplementary Material (*Supplementary files 1* and *2*); these can be applied to future multi-omics analyses of gene regulation and disease genetics.

# Materials and methods

## Key resources table

| Reagent type (species) or resource | Designation | Source or reference | Identifiers | Additional information |
|---|---|---|---|---|
| Cell line (*H. sapiens*, Male) | H19101 iPSC | 10.1101/gr.224436.117 | | |
| Antibody | Anti-acetyl Histone H3 (Lys27) (mouse monoclonal) | Wako Chemicals (USA) | 306–34849 | H3K27ac ChIP-seq |
| Antibody | Anti-cardiac troponin T (mouse monoclonal) | BD Biosciences | 564767 | CM flow cytometry |
| Chemical compound, drug | ROCK Y-27632 dihydrochloride | Abcam | ab120129, 10 mg | iPSC tissue culture |
| Chemical compound, drug | CHIR-99021 trihydrochloride | Tocris | 4953 | CM differentiation |
| Chemical compound, drug | Wnt-C59 | Tocris | 5148 | CM differentiation |
| Commercial assay or kit | TruSeq RNA libarary prep kit V2 | Illumina | RS-122–2001 | RNA-seq |
| Commercial assay or kit | NEBNext Multiplex Oligos for Illumina | NEB | E7335S | Hi-C |
| Commercial assay or kit | MEGAshortscript T7 Transcription Kit | Thermo Fisher | AM135 | Probe generation |
| Sequence-based reagent | Primer A | IDT | 5′-CTGGGAATCGCACCAGCGTGT-3′ | Probe generation |
| Sequence-based reagent | Primer B | IDT | 5′-CGTGGATGAGGAGCCGCAGTG-3′ | Probe generation |
| Sequence-based reagent | Primer A T7 | IDT | 5′-GGATTCTAATACGACTCACT ATAGGGATCGCACCAGCGTGT-3′ | Probe generation |
| Sequence-based reagent | blocking primer P5 | IDT | 1016184 | Hi-C capture |
| Sequence-based reagent | blocking primer P7 | IDT | 1016186 | Hi-C capture |

## Tissue culture of iPSCs

We used the Yoruban iPSC line 19101, kindly provided by the laboratory of Yoav Gilad. This iPSC line was reprogrammed from lymphoblastoid cells as part of a previous study, where it was shown to differentiate into all three germ layers, displayed a normal karyotype, and expressed markers characteristic of pluripotency (*Banovich et al., 2018*). iPSCs were grown in Essential 8 (E8) Medium (Thermo Fisher #A1517001) supplemented with 1X Penicillin-Streptomycin (Pen/Strep, Gibco) on Matrigel-coated tissue culture dishes (Corning #354277). Cells were passaged when they were ~80% confluent using enzyme-free dissociation solution (30 mM NaCl, 0.5 mM EDTA, 1X PBS minus Magnesium and Calcium) and maintained in E8 Medium with 10 μM Y-27632 dihydrochloride (Abcam #ab120129) for 24 hr. Medium was replaced daily. iPSC cultures routinely tested negative for mycoplasma contamination using the Universal Mycoplasma Detection Kit (ATCC #30–1012K).

## Cardiomyocyte differentiation

Cardiomyocyte differentiations were based on the protocol of *Burridge et al. (2014*) with modifications described in Banovich et al. (*Banovich et al., 2018*). iPSCs were expanded in 60 mm dishes in E8 media until they reached 60–70% confluency at which time the differentiation was started (day 0). On day 0, E8 media was replaced with 10 mL of basic heart media/12 μM GSK-3 inhibitor CHIR-99021 trihydrochloride (Tocris #4953)/Matrigel overlay [basic heart media: RPMI 1640 minus L-glutamine (HyClone #SH30096.01) with 1X GlutaMax (Life Technologies #11879020) supplemented with 1X B27 minus insulin (Thermo Fisher #A1895601) and 1X Pen/Strep; Matrigel overlay was accomplished by dissolving Matrigel in 50 mL basic heart media at a concentration of 0.5X according to

the lot-specific dilution factor]. After 24 hr (day 1), the GSK-3 inhibitor was removed by replacing media with 10 mL basic heart media. On day 3, media was replaced with 10 mL basic heart media supplemented with 2 µM Wnt-C59 (Tocris #5148). On day 5 (48 hours later), media was replaced with 10 mL basic heart media. On day 7, cells were washed once with 1X PBS and then 15 mL basic heart media was added. Media was replaced every other day in this way until day 15 at which time cardiomyocytes were selected for by replacing basic heart media with 10 mL lactate media (RPMI 1640 minus D-glucose, plus L-glutamine (Life Technologies #11879020), supplemented with 0.5 mg/mL recombinant human albumin (Sigma 70024-90-7), 5 mM sodium DL-lactate (Sigma 72-17-3), 213 µg/mL L-ascorbic acid 2-phosphate (Sigma 70024-90-7) and 1X Pen/Strep). Lactate media was replaced every other day until day 20 at which point cardiomyocytes were harvested. Cells from successful differentiations exhibited spontaneous beating around days 7–10.

Cardiomyocytes were harvested by washing once with 1X PBS followed by incubation in 4 mL TrypLE (Life Technologies 12604–021) at 37°C for 5 min. After incubation, 4 mL lactate media was added to the TrypLE and a 1 mL pipet was used to dislodge cells. Cells were strained once with a 100 µM strainer and then once with a 40 µM strainer. Cells were pelleted at 500xg and then resuspended in PBS and counted. For each batch of differentiation, 5 million cells were taken for promoter-capture Hi-C and 1 million cells were taken for RNA-seq. To assess purity, 2 million cells were taken for flow cytometry analysis using an antibody for cardiac Troponin T (BD Biosciences 564767). All cells used in downstream experiments were at least 86% Troponin T positive (*Figure 1—figure supplement 1A*). We carried out three independent differentiations of the same iPSC line and generated promoter-capture Hi-C and RNA-seq libraries in iPSCs and CMs from each triplicate.

## Promoter capture Hi-C
### Crosslinking cells
iPSCs or cardiomyocytes were harvested from tissue culture dishes and counted. Cells were resuspended in 1X PBS at a concentration of 1 million cells/mL and 37% formaldehyde was added to a final concentration of 1%. Crosslinking was carried out for 10 min at room temperature on a rocking platform. Glycine was added to a final concentration of 0.2 M to quench the reaction. The cells were pelleted, snap frozen in liquid nitrogen and stored at −80°C until ready for Hi-C processing.

### in situ Hi-C
We prepared all promoter capture Hi-C libraries in one batch using three crosslinked pellets of 5 million cells for both iPSCs and iPSC-derived cardiomyocytes, representing three independent cardiomyocyte differentiations. The in situ Hi-C step was performed as in *Rao et al. (2014)* with a single modification in which NEBNext reagents from the NEBNext Multiplex Oligos for Illumina kit were used (NEB #E7335S) instead of Illumina adapters, following the manufacturer's instructions. Hi-C libraries were amplified directly off of T1 beads (Life Technologies #65602) using NEBNext primers and six cycles of PCR.

### Promoter capture – probe design and generation
Hi-C capture probes were designed to target four MboI restriction fragment ends (120 bp) near the TSS of protein coding RefSeq genes (*O'Leary et al., 2016*) mapped to hg19 in the UCSC Genome Browser (*Speir et al., 2016*). To select restriction fragments, we only kept MboI restriction fragments longer than 200 bp and overlapping 10 kb around a RefSeq TSS. For TSSs closer than 1 kb from each other, only one was retained, as their interactions were likely to be captured by the other RefSeq TSS. The four MboI restriction fragment ends closest to each RefSeq TSS were selected as putative probes. The 120 bp sequences were submitted to Agilent's SureDesign proprietary software for probe selection, which can slightly shift the location and remove probes. In total, we ordered a library of 77,476 single-stranded DNA oligos from CustomArray, Inc. (www.customarrayinc.com). Each oligo consisted of the sequence 5′-ATCGCACCAGCGTGT$N_{120}$CACTGCGGCTCCTCA-3′ (*Gnirke et al., 2009*) where $N_{120}$ represents the 120 nucleotides adjacent to the MboI cut site. The complete list of oligo probes and their corresponding gene name is provided in *Supplementary file 9.1*.

The oligos arrived as a pool containing 1000 ng of material. We used 16 ng of the oligo pool in a PCR reaction to make them double stranded using primers 5′-CTGGGAATCGCACCAGCGTGT-3′

(Primer A), and 5′-CGTGGATGAGGAGCCGCAGTG-3′ (Primer B) as in (*Gnirke et al., 2009*). The PCR reaction was cleaned using AMPure XP beads (Agencourt #A6388) and eluted with 20 µl of water. To add the full T7 promoter to the 5′ end of the oligos, a second PCR reaction was carried out using 10 ng of the cleaned-up first-round PCR product with the forward primer 5′-GGATTCTAATACGACTCACTATAGGGATCGCACCAGCGTGT-3′ (Primer A T7). We purified the PCR product corresponding to 176 bp using a Qiagen gel extraction kit (#28704). To generate biotinylated RNA baits, we performed in vitro transcription on the double-stranded library using the MEGAshortscript T7 Transcription Kit (Thermo Fisher #AM135) with Biotin-16-dUTP (Sigma #11388908910). After DNase treatment the transcription reaction was cleaned using the MEGAclear kit (Thermo Fisher #AM1908) and eluted with 50 µl elution buffer. We confirmed the correct bait size on a denaturing gel.

## Promoter capture – hybridization with Hi-C library

To isolate promoter-containing fragments from the whole-genome in situ Hi-C library, we hybridized the biotinylated RNA bait pool with the Hi-C library as follows. A mix containing 500 ng of the Hi-C library, 2.5 µg of human Cot-1 DNA (Invitrogen #15279–011), 2.5 µg of salmon sperm DNA (Invitrogen #15632–011), 0.5 µl blocking primer P5 (IDT #1016184), and 0.5 µl blocking primer P7 (IDT #1016186) was heated for 5 min. at 95°, held at 65° and mixed with 13 µl pre-warmed hybridization buffer (10X SSPE, 10X Denhardt's, 10 mM EDTA and 0.2% SDS) and a 6 µl pre-warmed mix of 500 ng of the biotinylated RNA bait and 20U SUPERase-In (Thermo Fisher #AM2694). The hybridization mix was incubated for 24 hr at 65°C. To isolate captured fragments, we prepared 500 ng of streptavidin-coated magnetic beads (Dynabeads MyOne Streptavidin T1, Thermo Fisher #65601) in 200 µl of Binding buffer (1M NaCl, 10 mM Tris-HCl pH 7.5, 1 mM EDTA). The hybridization mix was added to the Streptavidin beads and rotated for 30 min at room temperature. The beads containing the captured Hi-C fragments were washed with 1X SSC, 0.1% SDS for 15 min at room temperature, followed by three washes (10 min each) at 65°C with 0.1X SSC/0.1% SDS. After the final wash, the beads were resuspended in 22 µl of water and proceeded to post-capture PCR. The PCR reaction was performed as before, with 11 µl of the 'capture Hi-C beads' and 8 cycles of amplification. An AMPure XP bead purification was used to clean the PCR reaction and DNA was quantified using the QuantiFluor dsDNA System (Promega #E2670) and a High Sensitivity Bioanalyzer. Final capture Hi-C libraries were subjected to 100 bp paired-end sequencing on an Illumina HiSeq 4000 machine. Read count summaries are provided in *Supplementary file 9.2*.

## Interaction calling

We used HiCUP v0.5.9 (*Wingett et al., 2015*) to align and filter Hi-C reads (total and filtered read counts are presented in *Supplementary file 9.2*). Unique reads were given to CHiCAGO version 1.2.0 (*Cairns et al., 2016*) and significant interactions were called with default parameters. In this study, we focused exclusively on *cis*-interactions as the evidence that *trans*-chromosomal interactions contribute to gene expression regulation is limited. CHiCAGO reports interactions for each captured restriction fragment; to summarize interactions by gene, we considered the interval spanning all captured fragments (i.e. the set of probes spanning each TSS) as the promoter region ('merged TSS'). This means the promoter regions created have variable lengths. In cases where multiple genes were annotated to the same promoter region, we report the interaction for each gene individually. This annotation allowed us to perform gene-level analyses, for example based on expression level. We removed this redundancy as necessary, for example in motif enrichment analyses of the promoter-interacting fragments. Using the 'merged TSS' interaction files, we filtered interactions to retain those that mapped within 1 kb of each other in at least two replicates. Specifically, we extended each promoter-interacting fragment by 1 kb on each end and then used BEDTools (*Quinlan and Hall, 2010*) pairToPair functionality to identify interactions where both ends matched across replicates. To identify cell type-specific interactions, we required that the interaction (with the 1 kb extension) was not present in any of the three replicates of the other cell type. The number of read-pairs per promoter and the corresponding number of significant interactions identified is presented in *Supplementary file 9.3*. The TAD analyses, motif enrichment, ChIP-seq peak enrichment, and eQTL analyses (related to *Figures 1*, *2*, *3* and *5*) were conducted with fragment-level interactions (no 1 kb

extension). The GWAS SNP analyses were conducted with 1kb-extended interactions, as we aimed to be as inclusive as possible when linking CVD SNPs to target genes.

PCHi-C interactions, TADs, RNA-seq, publicly available ChIP-seq, and GWAS SNPs are hosted by the WashU EpiGenome Browser (*Zhou et al., 2015*) as a public track hub. This can be accessed by going to http://epigenomegateway.wustl.edu/browser/. The public hub ('A promoter interaction map for cardiovascular disease genetics') can be found under the Human Hg19 browser.

## 4C-style plots

To generate the by-gene read counts displayed in the genome-browser figures, all read-pairs mapping to captured MboI fragments for a given promoter were summed across replicates. Specifically, we summed reads for each MboI fragment where the read was part of a paired-read that mapped to a bait for the given gene. The arcs that are displayed underneath the 4C-style plot represent significant interactions that were identified in at least two replicates as detailed above in 'Interaction calling'.

## TAD analysis

To identify TADs, we pooled reads across replicates for each cell type using the pre-capture Hi-C data (600M reads for iPSC and 733M reads for CM) and used HiCUP v0.5.9 (*Wingett et al., 2015*) to align and filter Hi-C reads. HOMER v4.8.3 (*Heinz et al., 2010*) was used to generate normalized interaction matrices at a resolution of 40 kb and then TopDom v0.0.2 (*Shin et al., 2016*) was used with a window size w = 10 to identify topological domains, boundaries and gaps. We only considered domains for the analyses in this paper. We considered a promoter capture Hi-C interaction to be 'intra-TAD' if the entire span of the interaction was fully contained in a single domain. 'Inter-TAD' interactions are defined as interactions where each end maps to a different domain.

## A/B compartments

The program runHiCpca.pl from the HOMER (*Heinz et al., 2010*) v4.8.3 package was used to call A/B compartments with -res 50000 for both whole-genome and capture Hi-C data.

## RNA-seq

Total RNA was extracted from flash-frozen pellets of 1 million cells using TRI Reagent (Sigma #T9424) and a homogenizer followed by RNA isolation and clean-up using the Direct-zol RNA Kit (Zymo Research #11–331). RNA-seq libraries were generated with the Illumina TruSeq V2 kit (Illumina, RS-122–2001) and 1 µg of RNA, following manufacturer's instructions. Libraries were made from RNA isolated from three independent iPSC-CM differentiations (triplicates of iPSC and of cardiomyocytes). Libraries were sequenced on an Illumina HiSeq 4000.

Gene counts were quantified with Salmon 0.7.2 (*Patro et al., 2017*) and imported with tximport 1.2.0 (*Soneson et al., 2015*) into DESeq2 1.12.4 (*Love et al., 2014*) to call differentially expressed genes. A minimum 1.5-fold-difference between CMs and iPSC triplicates and a minimum adjusted p-value of 0.05 were required to select differentially expressed genes for downstream analyses. TPMs (transcripts per million) were also estimated by Salmon. Because the samples clearly clustered according to their known tissues of origin (*Figure 1—figure supplement 2A*), no correction for batch effects was performed.

## H3K27ac ChIP-seq for comparison with epigenome roadmap samples

We performed ChIP-seq on 2.5 million cells each for iPSCs and CMs using H3K27ac antibodies (Wako #306–34849). Briefly, cells were crosslinked with 1% formaldehyde for 10 min at room temperature, quenched with 0.2M glycine for 5 min, pelleted and snap-frozen in liquid nitrogen. Cells were lysed in Lysis Buffer 1 (50 mM HEPES-KOH, pH 7.5, 140 mM NaCl, 1 mM EDTA, 10% glycerol, 0.5% NP-40, 0.25% Triton X-100). Crosslinked chromatin was sheared to an average size of 300 bp using a Bioruptor with 30" on/30" off at high setting and then incubated overnight at 4°C with 1 µg antibody. Dynabeads M-280 Sheep Anti-Mouse IgG (ThermoFisher #11201D) were used to pull down chromatin and ChIP DNA was eluted and prepared for sequencing using the NEBNext Ultra II DNA Library prep kit (NEB #E7645S). ChIP-seq reads were aligned with Bowtie 2–2.2.3 (*Langmead and Salzberg, 2012*) and peaks were called with HOMER (*Heinz et al., 2010*) v4.8.3 on

unique reads with mapping quality >10 using the –region and –style histone parameters. Significant peaks were overlapped with H3K27ac peaks from Epigenome Roadmap samples which demonstrated high concordance between matched tissue types (*Figure 1—figure supplement 2C,D*). Because we performed a low level of sequencing, we did not identify as many peaks as the Roadmap samples. Therefore, we used Roadmap ChIP-seq data in all of our analyses.

## Gene Ontology analysis

The human Gene Ontology (GO) associations of GO terms (*Ashburner et al., 2000*) to genes and the GO database were downloaded on January 22, 2016 from http://geneontology.org/gene-associations. GO terms were associated with RefSeq genes via gene symbols. Using the GO annotation graph, all parent terms were assigned to the terms annotated to a gene. A hypergeometric test was used to calculate the statistical significance of the difference of the number of genes associated with a given GO term in a particular gene set and the universe of all RefSeq genes (p<0.05). p-Values were corrected with the R package p.adjust function using the 'fdr' method.

For two of the GWAS disease groups (heart failure and myocardial infarction), the list of genes looping to LD SNPs included many histone genes. This is because there is a tag SNP located in the middle of a histone gene cluster (containing >30 histone genes located close together) in each case. After expanding the tag SNP to all SNPs in LD, many of the histone genes in that cluster looped to the LD SNPs, resulting in a high representation of these genes in the final gene list. The resulting Gene Ontology enrichment analysis gave terms relating to nucleosome and chromatin organization because of this over-representation. We therefore chose to remove these genes from the final gene lists of heart failure and myocardial infarction target genes.

## Motif analysis

The program findMotifsGenome.pl from the HOMER (*Heinz et al., 2010*) v4.8.3 package was used with –size given parameter to identify overrepresented motifs in the distal (non-promoter) interacting sequences of promoter interactions. As stated above, this analysis was performed on fragment-level promoter-interacting sequences.

## Histone ChIP-seq enrichment analysis

We obtained publicly available ChIP-seq data in the form of processed peak calls for H3K27ac, H3K4me1 and H3K27me3 from the Roadmap Epigenomics Project (*Kundaje et al., 2015*), and for CTCF from ENCODE (*ENCODE Project Consortium, 2012*) (*Supplementary file 10*). We only considered peaks that mapped outside of the captured region of promoters to ensure our results were not driven by the strong peak signal over most promoters. As a proxy for iPSCs, we used data from the H1 embryonic stem cell line and for CMs we used data from Left Ventricle tissue. We grouped genes into five expression categories based on the average TPM values: group 1 (0 TPM), group 2 (TPM 0–3), group 3 (TPM 3–25), group 4 (TPM 25–150) and group 5 (TPM >150) and for each group of genes, we calculated the enrichment for promoter interactions to overlap a given feature. To calculate enrichment of interactions overlapping an epigenetic feature, we compared the observed proportion of MboI fragments in significant interactions overlapping a feature to the proportion of random MboI fragments overlapping the feature. Specifically, we randomly selected MboI fragments from a set that excluded fragments mapping within captured regions (promoters) or within unmappable genomic regions (gaps). The number of randomly selected fragments matched the number of interacting fragments considered for the analysis. We performed 100 iterations of overlapping random fragments with a feature and report the average fold-enrichment. We refer to this method of enrichment as a 'genome-wide' background model because for each gene expression group, the observed proportion of fragments containing a peak is compared to randomly selected fragments from the whole genome.

To calculate the correlation between expression and histone ChIP-seq peak density, we calculated the Spearman's rank correlation between the expression value for each gene (the average TPM value) and the number of peaks mapping within 300 kb of each gene TSS. We only considered genes with at least one significant interaction in the respective cell type to allow for generalizations to the enrichment analysis presented in *Figure 3*.

## GWAS analysis

We compiled genome-wide significant SNPs associated with GWAS for cardiac arrhythmia, heart failure, and myocardial infarction from the NHGRI-EBI database (http://www.ebi.ac.uk/gwas/); see *Supplementary file 7* for list of terms used to identify specific GWAS. We expanded each set of SNPs to all SNPs in high LD ($r^2$ >0.9) using phase 3 data of the 1000 genomes project (*Nikpay et al., 2015*) (*Supplementary file 3*). For each lead SNP from the GWAS we analyzed, we selected a 100 kb interval centered on the SNP (SNP ± 50 kb). For each 100 kb interval, Tabix (*Li, 2011*) was used to retrieve genotypes. We then used PLINK (*Purcell et al., 2007*) v1.90p on phase three data from the 1000 genomes project (*Nikpay et al., 2015*) (ftp.1000genomes.ebi.ac.uk/vol1/ftp/release/20130502, v5a) to select SNPs in LD ($r^2$ >0.9) with the tag SNP and a minimum allele frequency of 0.01. We only included the populations primarily studied in the GWASs: CEU (central European), ASW (African American) and JPT (Japanese). We assigned all SNPs in promoter-distal interactions to their interacting gene(s) ('target genes') using cardiomyocyte promoter capture Hi-C data. We did not require the SNP to map to regions associated with open chromatin or enhancer marks as these types of data are highly cell-type specific and we did not wish to exclude SNPs in regions that may be active in non-assayed cell types.

We note that one major GWAS for dilated cardiomyopathy was not included in the NHGRI-EBI database (*Meder et al., 2014*), likely because there is an error obtaining the online methods of the paper. After careful inspection of the study, we concluded that the GWAS met the NHGRI-EBI criteria and included the associations from that study in our analysis. A complete list of all studies used in this analysis can be found in *Supplementary file 8*.

## MGI analysis

To calculate enrichment of target genes to cause cardiovascular phenotypes when deleted in mice (Mouse Genome Informatics database), we randomly selected 347 genes from the list of starting genes (i.e. genes with at least one promoter-distal interaction in CMs, meaning it could be a target gene), and calculated the proportion that caused a cardiovascular phenotype in mice. We performed this randomized selection for 1000 iterations to generate the randomized (expected) values. Random genes were not required to be expressed, as the set of target genes contains genes that are not expressed. p-Value was calculated with a Z test.

## eQTL analysis

For eQTLs used in comparisons with GWAS variants and Hi-C interactions, we used the set of GTEx v7 eQTLs identified as significant in the left ventricle of the heart (*Carithers et al., 2015*). eQTLs were called significant if q < 0.05 after false discovery rate correction (*Storey and Tibshirani, 2003*). We only considered promoter-distal eQTLs that were at least 10 kb from their associated gene to allow for that eQTL to map to an interaction with it's associated gene.

To calculate enrichment for eQTLs to loop to their associated gene, we used a background model whereby each promoter's set of interactions were re-mapped to a different promoter, keeping the distance and strand orientation consistent. We performed this re-mapping of all promoter interactions 1000 times and calculated the proportion of all eQTLs that mapped to interactions for their eQTL-associated gene in each permutation. We either used the CM interactions or the iPSC interactions with the same set of left ventricle eQTLs to compare cell-type specificity of the promoter interaction data.

## Data availability

Raw and processed sequencing data are provided at ArrayExpress through accession numbers E-MTAB-6014 (Hi-C) and E-MTAB-6013 (RNA-seq).

## Acknowledgements

We kindly thank the laboratory of Yoav Gilad for providing the iPSC line and assisting with the cardiomyocyte differentiation protocol, and Dr. Kohta Ikegami for assistance with the ChIP-seq protocol. This work was supported by NIH grants HL123857 (MAN), HL119967 (MAN), HL118758

(MAN), HL128075 (MAN and EMM), T32GMOO7197 (LEM), American Heart Association Pre-doctoral award 17PRE33410726 (LEM), HL137307 (LEM).

## Additional information

### Funding

| Funder | Grant reference number | Author |
|---|---|---|
| National Institutes of Health | HL123857 | Marcelo A Nóbrega |
| National Institutes of Health | HL119967 | Marcelo A Nóbrega |
| National Institutes of Health | HL118758 | Marcelo A Nóbrega |
| National Institutes of Health | HL128075 | Elizabeth M McNally Marcelo A Nóbrega |
| National Institutes of Health | T32GMOO7197 | Lindsey E Montefiori |
| American Heart Association | 17PRE33410726 | Lindsey E Montefiori |
| National Institutes of Health | HL137307-01 | Lindsey E Montefiori |

The funders had no role in study design, data collection and interpretation, or the decision to submit the work for publication.

### Author contributions

Lindsey E Montefiori, Formal analysis, Methodology, Writing—original draft; Debora R Sobreira, Ivy Aneas, Amelia C Joslin, Grace T Hansen, Grazyna Bozek, Methodology, Writing—review and editing; Noboru J Sakabe, Formal analysis, Supervision, Writing—review and editing; Ivan P Moskowitz, Elizabeth M McNally, Supervision, Writing—review and editing; Marcelo A Nóbrega, Conceptualization, Supervision, Funding acquisition, Writing—original draft, Project administration, Writing—review and editing

### Author ORCIDs

Lindsey E Montefiori (iD) http://orcid.org/0000-0003-2342-6349
Ivan P Moskowitz (iD) http://orcid.org/0000-0003-0014-4963
Marcelo A Nóbrega (iD) http://orcid.org/0000-0002-0451-7846

### Decision letter and Author response

Decision letter https://doi.org/10.7554/eLife.35788.102
Author response https://doi.org/10.7554/eLife.35788.103

## Additional files

### Supplementary files

• Supplementary file 1. PCHi-C interactions for iPSC. Significant interactions in iPSC (identified in at least two out of three replicates) are presented in paired bed format. Column seven is the CHiCAGO score; Column eight contains the gene information (gene name, identifier, strand, TSS position). If the interaction involves another promoter, the second gene name information is provided.
DOI: https://doi.org/10.7554/eLife.35788.019

• Supplementary file 2. PCHi-C interactions for CM. Significant interactions in CM (identified in at least two out of three replicates) are presented in paired bed format. Column seven is the CHiCAGO score; Column eight contains the gene information (gene name, identifier, strand, TSS position). If the interaction involves another promoter, the second gene name information is provided.
DOI: https://doi.org/10.7554/eLife.35788.020

• Supplementary file 3. CVD SNPs. All SNPs in high LD ($r^2 > 0.9$) with CVD tag SNPs are provided. The first four columns indicate the tag SNP position; columns 5–8 indicate the SNPs in LD with the tag SNP; column nine is the degree of LD ($r^2$ value).

DOI: https://doi.org/10.7554/eLife.35788.021

• Supplementary file 4. HOMER motif analysis for the distal interacting regions of promoter interactions. For each analysis presented in *Figure 2*, the full output of motifs identified in interacting fragments are listed.
DOI: https://doi.org/10.7554/eLife.35788.022

• Supplementary file 5. Gene Ontology enrichment output. The full output of the GO enrichment analysis is provided.
DOI: https://doi.org/10.7554/eLife.35788.023

• Supplementary file 6. Gene Ontology input gene lists. The list of genes used as input for GO analysis is provided.
DOI: https://doi.org/10.7554/eLife.35788.024

• Supplementary file 7. GWAS terms used to compile studies. The list of trait terms used to filter GWAS studies is provided.
DOI: https://doi.org/10.7554/eLife.35788.025

• Supplementary file 8. GWAS summary table. Table 8.1 contains information related to the CVD GWAS used in this paper, including PubMed ID, first author, date of publication, journal, study title, tag SNP chromosome position, rsID, trait. Table 8.2 contains information on each LD SNP-target gene interaction (tag SNP and corresponding LD SNP, target gene, interaction coordinates, target gene expression in iPSC and CM, MGI cardiovascular phenotype information.
DOI: https://doi.org/10.7554/eLife.35788.026

• Supplementary file 9. Hi-C read information. Table 9.1 contains the probe sequences used for promoter capture, along with the corresponding gene name. Table 9.2 contains the total number of sequenced and processed/filtered reads for each Hi-C experiment. Table 9.3 contains the number of reads mapping to each promoter and the corresponding number of significant (present in at least two replicates) interactions called.
DOI: https://doi.org/10.7554/eLife.35788.027

• Supplementary file 10. Public datasets used. ChIP-seq and RNA-seq datasets used in our analyses are listed.
DOI: https://doi.org/10.7554/eLife.35788.028

• Transparent reporting form
DOI: https://doi.org/10.7554/eLife.35788.029

### Data availability

Raw and processed sequencing data are provided at ArrayExpress through accession numbers E-MTAB-6014 (Hi-C) and E-MTAB-6013 (RNA-seq).

The following datasets were generated:

| Author(s) | Year | Dataset title | Dataset URL | Database, license, and accessibility information |
|---|---|---|---|---|
| Montefiori LE, Sobreira DR, Sakabe NJ, Aneas I, Joslin AC, Hansen GT, Bozek G, Moskowitz IP, McNally EM, Nóbrega MA | 2018 | Capture Hi-C in iPSC and CM | https://www.ebi.ac.uk/arrayexpress/experiments/E-MTAB-6014/ | Publicly available at ArrayExpress (accession no. E-MTAB-6014) |
| Montefiori LE, Sobreira DR, Sakabe NJ, Aneas I, Joslin AC, Hansen GT, Bozek G, Moskowitz IP, McNally EM, Nóbrega MA | 2018 | RNA-seq in iPSC and CM | https://www.ebi.ac.uk/arrayexpress/experiments/E-MTAB-6013/ | Publicly available at ArrayExpress (accession no. E-MTAB-6013) |

The following previously published datasets were used:

| | | | | Database, license, |
|---|---|---|---|---|

| Author(s) | Year | Dataset title | Dataset URL | and accessibility information |
|---|---|---|---|---|
| Kundaje A | 2015 | Roadmap Epigenome Project - H1-H3K4me1 | http://egg2.wustl.edu/roadmap/data/byFile-Type/peaks/consoli-dated/narrowPeak/E003-H3K4me1.narrowPeak.gz | Open public database: Roadmap Epigenome Project |
| Kundaje A | 2015 | Roadmap Epigenome Project-H1-H3K27ac | http://egg2.wustl.edu/roadmap/data/byFile-Type/peaks/consoli-dated/narrowPeak/E003-H3K27ac.narrowPeak.gz | Open public database: Roadmap Epigenome Project |
| Kundaje A | 2015 | Roadmap Epigenome Project-H1-H3K27me3 | http://egg2.wustl.edu/roadmap/data/byFile-Type/peaks/consoli-dated/narrowPeak/E003-H3K27me3.narrowPeak.gz | Open public database: Roadmap Epigenome Project |
| Kundaje A | 2015 | Roadmap Epigenome Project-H9-H3K4me1 | http://egg2.wustl.edu/roadmap/data/byFile-Type/peaks/consoli-dated/narrowPeak/E008-H3K4me1.narrowPeak.gz | Open public database: Roadmap Epigenome Project |
| Kundaje A | 2015 | Roadmap Epigenome Project-H9-H3K27ac | http://egg2.wustl.edu/roadmap/data/byFile-Type/peaks/consoli-dated/narrowPeak/E008-H3K27ac.narrowPeak.gz | Open public database: Roadmap Epigenome Project |
| Kundaje A | 2015 | Roadmap Epigenome Project-H9-H3K27me3 | http://egg2.wustl.edu/roadmap/data/byFile-Type/peaks/consoli-dated/narrowPeak/E008-H3K27me3.narrowPeak.gz | Open public database: Roadmap Epigenome Project |
| Kundaje A | 2015 | Roadmap Epigenome Project-LV-H3K4me1 | http://egg2.wustl.edu/roadmap/data/byFile-Type/peaks/consoli-dated/narrowPeak/E095-H3K4me1.narrowPeak.gz | Open public database: Roadmap Epigenome Project |
| Kundaje A | 2015 | Roadmap Epigenome Project-LV-H3K27ac | http://egg2.wustl.edu/roadmap/data/byFile-Type/peaks/consoli-dated/narrowPeak/E095-H3K27ac.narrowPeak.gz | Open public database: Roadmap Epigenome Project |
| Kundaje A | 2015 | Roadmap Epigenome Project-LV-H3K27me3 | http://egg2.wustl.edu/roadmap/data/byFile-Type/peaks/consoli-dated/narrowPeak/E095-H3K27me3.narrowPeak.gz | Open public database: Roadmap Epigenome Project |
| Kundaje A | 2015 | Roadmap Epigenome Project-RV-H3K4me1 | http://egg2.wustl.edu/roadmap/data/byFile-Type/peaks/consoli-dated/narrowPeak/E105-H3K4me1.narrowPeak.gz | Open public database: Roadmap Epigenome Project |
| Kundaje A | 2015 | Roadmap Epigenome Project-RV-H3K27ac | http://egg2.wustl.edu/roadmap/data/byFile-Type/peaks/consoli-dated/narrowPeak/E105-H3K27ac.narrowPeak.gz | Open public database: Roadmap Epigenome Project |
| Kundaje A | 2015 | Roadmap Epigenome Project-RV-H3K27me3 | http://egg2.wustl.edu/roadmap/data/byFile-Type/peaks/consoli-dated/narrowPeak/E105-H3K27me3.narrowPeak.gz | Open public database: Roadmap Epigenome Project |

| | | | | |
|---|---|---|---|---|
| Kundaje A | 2015 | Roadmap Epigenome Project-RA-H3K4me1 | http://egg2.wustl.edu/roadmap/data/byFile-Type/peaks/consoli-dated/narrowPeak/E104-H3K4me1.narrowPeak.gz | Open public database: Roadmap Epigenome Project |
| Kundaje A | 2015 | Roadmap Epigenome Project-RA-H3K27ac | http://egg2.wustl.edu/roadmap/data/byFile-Type/peaks/consoli-dated/narrowPeak/E104-H3K27ac.narrowPeak.gz | Open public database: Roadmap Epigenome Project |
| Kundaje A | 2015 | Roadmap Epigenome Project-RA-H3K27me3 | http://egg2.wustl.edu/roadmap/data/byFile-Type/peaks/consoli-dated/narrowPeak/E104-H3K27me3.narrowPeak.gz | Open public database: Roadmap Epigenome Project |
| ENCODE Project Consortium | 2012 | H1-CTCF | https://www.ncbi.nlm.nih.gov/geo/query/acc.cgi?acc=GSE29611 | Publicly available at the NCBI Gene Expression Omnibus (accession no: GSE29611) |
| ENCODE Project Consortium | 2012 | Human CM - CTCF | https://www.ncbi.nlm.nih.gov/geo/query/acc.cgi?acc=GSE35583 | Publicly available at the NCBI Gene Expression Omnibus (accession no: GSE35583) |
| Bernstein BE, Sta-matoyannopoulos JA, Costello JF, Ren B | 2010 | ENCODE_RNA-seq-fetal heart 1 | https://www.ncbi.nlm.nih.gov/sra/SRR643778/ | Publicly available at the NCBI Gene Expression Omnibus |
| Bernstein BE, Sta-matoyannopoulos JA, Costello JF, Ren B | 2010 | ENCODE_RNA-seq-fetal heart 2 | https://www.ncbi.nlm.nih.gov/sra/SRR643779/ | Publicly available at the NCBI Gene Expression Omnibus |
| Bernstein BE, Sta-matoyannopoulos JA, Costello JF, Ren B | 2010 | ENCODE_RNA-seq-H1-1 | https://www.ncbi.nlm.nih.gov/geo/query/acc.cgi?acc=GSM438361 | Publicly available at the NCBI Gene Expression Omnibus (accession no: GSM438361) |
| Bernstein BE, Sta-matoyannopoulos JA, Costello JF, Ren B | 2010 | ENCODE_RNA-seq-H1-2 | https://www.ncbi.nlm.nih.gov/geo/query/acc.cgi?acc=GSM958737 | Publicly available at the NCBI Gene Expression Omnibus (accession no: GSM958737) |
| Bernstein BE, Sta-matoyannopoulos JA, Costello JF, Ren B | 2010 | ENCODE_RNA-seq-H1-4 | https://www.ncbi.nlm.nih.gov/geo/query/acc.cgi?acc=GSM1817053 | Publicly available at the NCBI Gene Expression Omnibus (accession no: GSM1817053) |
| Bernstein BE, Sta-matoyannopoulos JA, Costello JF, Ren B | 2010 | ENCODE_RNA-seq-LV-1 | https://www.ncbi.nlm.nih.gov/geo/query/acc.cgi?acc=GSM1010938 | Publicly available at the NCBI Gene Expression Omnibus (accession no: GSM10 10938) |
| Bernstein BE, Sta-matoyannopoulos JA, Costello JF, Ren B | 2010 | ENCODE_RNA-seq-LV-2 | https://www.ncbi.nlm.nih.gov/geo/query/acc.cgi?acc=GSM1010964 | Publicly available at the NCBI Gene Expression Omnibus (accession no: GSM10 10964) |
| Kilpinen H, Waszak SM, Gschwind AR, Raghav SK, Wit-wicki RM, Orioli A, Migliavacca E, Wiederkehr M, Gu-tierrez-Arcelus M, Panousis N, Yur- | 2013 | 1000G_RNA-seq-LCL_1883_GM11830_1 | ftp://ftp.sra.ebi.ac.uk/vol1/fastq/ERR356/ERR356375/ERR356375_1.fastq.gz | Publicly available at ArrayExpress |

| | | | | |
|---|---|---|---|---|
| ovsky A, Lappalainen T, Romano-Palumbo L, Planchon A, Bielser D, Bryois J, Padioleau I, Udin G, Thurnheer S, Hacker D, Core LJ, Lis JT, Hernandez N, Reymond A, Deplancke B, Dermitzakis ET | | | | |
| Kilpinen H, Waszak SM, Gschwind AR, Raghav SK, Witwicki RM, Orioli A, Migliavacca E, Wiederkehr M, Gutierrez-Arcelus M, Panousis N, Yurovsky A, Lappalainen T, Romano-Palumbo L, Planchon A, Bielser D, Bryois J, Padioleau I, Udin G, Thurnheer S, Hacker D, Core LJ, Lis JT, Hernandez N, Reymond A, Deplancke B, Dermitzakis ET | 2013 | 1000G_RNA-seq-LCL_1883_GM11894_1 | ftp://ftp.sra.ebi.ac.uk/vol1/fastq/ERR356/ERR356368/ERR356368_1.fastq.gz | Publicly available at ArrayExpress |
| Kilpinen H, Waszak SM, Gschwind AR, Raghav SK, Witwicki RM, Orioli A, Migliavacca E, Wiederkehr M, Gutierrez-Arcelus M, Panousis N, Yurovsky A, Lappalainen T, Romano-Palumbo L, Planchon A, Bielser D, Bryois J, Padioleau I, Udin G, Thurnheer S, Hacker D, Core LJ, Lis JT, Hernandez N, Reymond A, Deplancke B, Dermitzakis ET | 2013 | 1000G_RNA-seq-LCL_1883_GM12043_1 | ftp://ftp.sra.ebi.ac.uk/vol1/fastq/ERR356/ERR356365/ERR356365_1.fastq.gz | Publicly available at ArrayExpress |
| Kilpinen H, Waszak SM, Gschwind AR, Raghav SK, Witwicki RM, Orioli A, Migliavacca E, Wiederkehr M, Gutierrez-Arcelus M, Panousis N, Yurovsky A, Lappalainen T, Romano-Palumbo L, Planchon A, Bielser D, Bryois J, Padioleau I, Udin G, Thurnheer S, Hacker D, Core LJ, Lis JT, Hernandez N, Reymond A, Deplancke B, Dermitzakis ET | 2013 | 1000G_RNA-seq-LCL_1883_GM12878_1 | ftp://ftp.sra.ebi.ac.uk/vol1/fastq/ERR356/ERR356372/ERR356372_2.fastq.gz | Publicly available at ArrayExpress |
| Kilpinen H, Waszak SM, Gschwind AR, Raghav SK, Witwicki RM, Orioli A, | 2012 | ENCODE_RNA-seq-LCL_encode_1 | https://www.encodeproject.org/files/ENCFF000CXI/@@download/ENCFF000CXI. | Publicly available at ArrayExpress |

| | | | | |
|---|---|---|---|---|
| Migliavacca E, Wiederkehr M, Gutierrez-Arcelus M, Panousis N, Yurovsky A, Lappalainen T, Romano-Palumbo L, Planchon A, Bielser D, Bryois J, Padioleau I, Udin G, Thurnheer S, Hacker D, Core LJ, Lis JT, Hernandez N, Reymond A, Deplancke B, Dermitzakis ET | | | fastq.gz | |
| Kilpinen H, Waszak SM, Gschwind AR, Raghav SK, Witwicki RM, Orioli A, Migliavacca E, Wiederkehr M, Gutierrez-Arcelus M, Panousis N, Yurovsky A, Lappalainen T, Romano-Palumbo L, Planchon A, Bielser D, Bryois J, Padioleau I, Udin G, Thurnheer S, Hacker D, Core LJ, Lis JT, Hernandez N, Reymond A, Deplancke B, Dermitzakis ET | 2012 | ENCODE_RNA-seq-LCL_encode_2 | https://www.encodeproject.org/files/ENCFF000CXH/@@download/ENCFF000CXH.fastq.gz | Publicly available at ArrayExpress |
| Kilpinen H, Waszak SM, Gschwind AR, Raghav SK, Witwicki RM, Orioli A, Migliavacca E, Wiederkehr M, Gutierrez-Arcelus M, Panousis N, Yurovsky A, Lappalainen T, Romano-Palumbo L, Planchon A, Bielser D, Bryois J, Padioleau I, Udin G, Thurnheer S, Hacker D, Core LJ, Lis JT, Hernandez N, Reymond A, Deplancke B, Dermitzakis ET | 2013 | GEUVADIS_RNA-seq-LCL_geuvadis_1 | ftp://ftp.sra.ebi.ac.uk/vol1/fastq/ERR187/ERR187488/ERR187488.fastq.gz | Publicly available at ArrayExpress |
| Kilpinen H, Waszak SM, Gschwind AR, Raghav SK, Witwicki RM, Orioli A, Migliavacca E, Wiederkehr M, Gutierrez-Arcelus M, Panousis N, Yurovsky A, Lappalainen T, Romano-Palumbo L, Planchon A, Bielser D, Bryois J, Padioleau I, Udin G, Thurnheer S, Hacker D, Core LJ, Lis JT, Hernandez N, Reymond A, Deplancke B, Dermitzakis ET | 2013 | GEUVADIS_RNA-seq-LCL_geuvadis_2 | ftp://ftp.sra.ebi.ac.uk/vol1/fastq/ERR187/ERR187490/ERR187490.fastq.gz | Publicly available at ArrayExpress |

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
