## [Decision Letter]

Thank you for submitting your article "A promoter interaction map for cardiovascular disease genetics" for consideration by *eLife*. Your article has been reviewed by three peer reviewers, and the evaluation has been overseen by a Reviewing Editor and Mark McCarthy as the Senior Editor. The following individual involved in review of your submission has agreed to reveal his identity: Daan Noordermeer (Reviewer #3).

The reviewers have discussed the reviews with one another and the Reviewing Editor has drafted this decision to help you prepare a revised submission.

The manuscript presents PCHi-C data for iPSCs and CMs with the goal to identify target genes of non-coding loci identified in GWAS. They find that for 19% of LD SNPs identified in GWAS they find significant long-range interactions with distal genes that are likely candidates for genes mis-regulated in CVD. The manuscript presents two main datasets: capture data for promoters in iPSC and CM, and also RNAseq and H3K27ac ChIP. The bulk of the paper is dedicated to showing that the iPSCs and CM cells are indeed what they are supposed to be, and to show that the capture Hi-C data is good. The last part of the paper is focused on showing that a set of significant interactions connect GWAS SNPs to putative target genes. This can be a useful resource for the community.

Essential revisions:

1) Throughout the paper you have shown arcs indicating significant interactions. These should be replaced, or complemented with tracks showing the real underlying data, e.g. "4C-style" plots.

2) Place significant interactions in the context of TADs: are these mostly within TADs or do they also connect loci in separate TADs? Do the significant interactions provide additional information over and above TAD-based enhancer-promoter predictions?

3) The analysis in Figure 3 needs to be re-evaluated. Whereas the authors claim positive correlations, the reported Spearman correlations ranging from 0.09 – 0.2 suggest a very weak trend at best, with a large degree of variation within each category (as confirmed by the box-plots). Very significant P-values are reported to support the correlation, but the approach based on genome-wide randomization of H3K27ac-peaks is not appropriate. If anything, the current analysis confirms the tendency of active genes and their regulatory elements to cluster together in the genome (see e.g. RIDGES (Caron et al., 2001) and Hi-C compartment A (Lieberman-Aiden et al., 2009)). Further, the claim that number of H3K27ac and H3K4me1 peaks surrounding a promoter alone is a good predictor of gene activity does not take into account which peaks are contacted by promoters, which may constitute a large fraction. To improve the analysis, the authors should take a similar approach as Schoenfelder et al., (2015; Figure 3H), where for different interaction counts the distribution of expression groups is compared. Subsequently, more relevant p-values can be determined by randomizing the expression groups for all genes (without changing the position of genes). Taking this approach, the authors should be able to get similar correlations for interaction numbers as reported by Schoenfelder et al., (r = 0.98), whereas it will be interesting to see if similar correlations can be obtained by looking at the number of surrounding H3K27ac peaks, taking into account if they are contacted or not.

4) The paper would be further strengthened by providing examples that some of the predicted long-range interactions are functional. Are any of the predicted target genes affected in disease, e.g. reduced or increased in expression in individuals carrying the distal variant?

---

## [Author Response]

The manuscript presents PCHi-C data for iPSCs and CMs with the goal to identify target genes of non-coding loci identified in GWAS. They find that for 19% of LD-SNPs identified in GWAS they find significant long-range interactions with distal genes that are likely candidates for genes mis-regulated in CVD. The manuscript presents two main datasets: capture data for promoters in iPSC and CM, and also RNAseq and H3K27ac ChIP. The bulk of the paper is dedicated to showing that the iPSCs and CM cells are indeed what they are supposed to be, and to show that the capture Hi-C data is good. The last part of the paper is focused on showing that a set of significant interactions connect GWAS SNPs to putative target genes. This can be a useful resource for the community.

We thank the reviewers for their positive assessment of our manuscript, and for providing helpful and critical feedback. We address all of the reviewer’s comments and concerns below. As we aim to provide the iPSC and CM maps as an accessible resource, we have hosted the full set of data presented in this study at the WashU EpiGenome Browser: http://epigenomegateway.wustl.edu/browser/?genome=hg19&session=X8F93vWJ7j&statusId=1128458586

Essential revisions:1) Throughout the paper you have shown arcs indicating significant interactions. These should be replaced, or complemented with tracks showing the real underlying data, e.g. "4C-style" plots.

We thank the reviewers for this suggestion. We have added a track that displays the read counts for all MboI fragments (summed across replicates) mapping to each promoter in the figures where we show interaction data. These counts represent the values CHiCAGO used to identify significant interactions (i.e. filtered reads).

2) Place significant interactions in the context of TADs: are these mostly within TADs or do they also connect loci in separate TADs? Do the significant interactions provide additional information over and above TAD-based enhancer-promoter predictions?

We analyzed our pre-capture Hi-C data in order to identify TADs and added a series of analyses that place promoter interactions in the context of TADs (Figure 1—figure supplement 3). We show that the majority of interactions are contained within TADs (73% in iPSCs and 77% in CMs), in line with what has been reported for other PCHi-C datasets (Freire-Pritchett et al., 2017). We characterized several differences between intra- and inter-TAD interactions, namely that inter-TAD interactions tend to be weaker as determined by the CHiCAGO score and are slightly more enriched for looping to CTCF sites compared to intra-TAD interactions. Furthermore, promoters with inter-TAD interactions tend to be located close to TAD boundaries and were generally higher expressed than promoters with intra-TAD interactions, particularly in cardiomyocytes. These observations are in line with known features of TAD boundaries, which are enriched for housekeeping genes (e.g. Dixon et al., 2012).

Although TADs represent relatively invariant and small (< 1Mb) units of *cis*-regulatory blocks, it is well established that dynamic, cell type-specific *cis*-regulatory interactions occur within TADs to direct cell differentiation (Dixon et al., 2012, Nora et al., 2012, Freire-Pritchett et al., 2017, Siersbaek et al., 2017, Dixon et al., 2015). Without high resolution promoter interaction data, which identifies precise sub-kilobase promoter contacts, or other similarly high-resolution approaches such as 5C, the within-TAD structure is obscured due to limitations in resolution of whole-genome Hi-C datasets. Even the most deeply sequenced Hi-C data (Rao et al., 2014) identified only ~10,000 loops which mostly represent structural interactions. It is within these structural (i.e. CTCF/cohesin-mediated) loops that enhancer-promoter (E-P) interactions are thought to occur. The more transient nature of E-P contacts necessitates a method that directly enriches for these interactions in Hi-C datasets.

For example, the TADs surrounding the *IRX5* locus appear similar in iPSCs and CMs, however PCHi-C clearly identifies E-P interactions between *IRX5* and a distal enhancer element specifically in CMs (Author response image 1). This dynamic profile of *IRX5* interactions is likely only to be fully realized with a promoter capture approach.

We attempted to quantify the challenge of identifying E-P interactions using only TAD calls by calculating the number of possible E-P connections within TADs. We considered only active genes (TPM of at least 1), non-promoter H3K27ac ChIP-seq peaks, and interactions that map fully within TADs. The average number of potential E-P interactions within TADs is 136 for iPSCs and 144 for CMs, compared to an observed average of only 5 E-P contacts per TAD in iPSCs and 7 in CMs from PCHi-C. While we acknowledge that PCHi-C data is still very sparse leading to many missed interactions, we believe that our data indicate that the number of realized functional interactions within TADs is likely restricted and necessitates direct assaying (through PCHi-C or other techniques) to identify E-P interactions in an otherwise complex landscape of many promoters and many enhancer elements.

**Author response image 1. respfig1:** *Cis*-regulation within TADs. Genome browser snapshot of the *IRX5* locus in iPSCs (top) and CMs (bottom). Yellow highlighted region is the *IRX5* promoter. CM-specific interactions to a Vista heart enhancer and H3K27ac peaks are highlighted in gray. Note the relatively invariant TAD structure over this region, compared to the dynamic within-TAD *IRX5* promoter interactions between the two cell types (black arrowheads).

3) The analysis in Figure 3 needs to be re-evaluated. Whereas the authors claim positive correlations, the reported Spearman correlations ranging from 0.09 – 0.2 suggest a very weak trend at best, with a large degree of variation within each category (as confirmed by the box-plots). Very significant P-values are reported to support the correlation, but the approach based on genome-wide randomization of H3K27ac-peaks is not appropriate. If anything, the current analysis confirms the tendency of active genes and their regulatory elements to cluster together in the genome (see e.g. RIDGES (Caron et al., 2001) and Hi-C compartment A (Lieberman-Aiden et al., 2009)).

We appreciate the reviewer’s criticism. We agree that our original presentation of the analysis in Figure 3 was suboptimal and we have fully revised the figure and the Results section to make the analysis and interpretation of the data more clear (see below).

Further, the claim that number of H3K27ac and H3K4me1 peaks surrounding a promoter alone is a good predictor of gene activity does not take into account which peaks are contacted by promoters, which may constitute a large fraction. To improve the analysis, the authors should take a similar approach as Schoenfelder et al., (2015; Figure 3H), where for different interaction counts the distribution of expression groups is compared. Subsequently, more relevant p-values can be determined by randomizing the expression groups for all genes (without changing the position of genes). Taking this approach, the authors should be able to get similar correlations for interaction numbers as reported by Schoenfelder et al., (r = 0.98), whereas it will be interesting to see if similar correlations can be obtained by looking at the number of surrounding H3K27ac peaks, taking into account if they are contacted or not.

In Figure 3H of Schoenfelder et al., 2015, the number of promoters looping to between 0 and >10 enhancer elements is depicted, with promoters grouped into 5 expression categories. The corresponding text that references this figure reports a Spearman correlation of 0.975 “between gene expression level and the number of interacting enhancer elements.” We confirmed with the authors that they obtained this value of 0.975 by correlating the median number of enhancer elements contacting promoters in each expression group with the expression group. We analyzed our data using the same approach and we also obtained very high Spearman’s correlations (Spearman’s rho = 0.89 in iPSC and 0.98 in CM, Author response image 2).

However, as can be seen from Author response image 2, the median number of enhancer elements contacting promoters in each expression group does not accurately reflect the distribution of numbers of enhancers contacting promoters per group, as there is a wide range across all expression groups. We reasoned that an alternative way to represent the relationship between expression and number of enhancer contacts is to perform the correlation using all data points (~13k) instead of 5. These are the original correlations that we reported (Spearman’s rho=0.09 for iPSC and 0.16 for CM, *P*<2.2x10^-16^). These correlations are low, but positive, and this reflects that the number of enhancer contacts only partially captures the logic underlying expression levels. A correlation of nearly 1 would suggest, misleadingly, that this is the case.

In our original Figure 3, we set out to explain why highly expressed genes are more enriched for looping to enhancer marks than lowly expressed genes. This trend has been reported previously (e.g. Schoenfelder et al., 2017, Javierre et al., 2017). We reasoned this increasing enrichment reflects the fact that highly expressed genes are located in genomic regions with more ChIP-seq peaks of open chromatin marks; additionally, highly expressed genes have more long-range interactions in general (i.e. the number of ChIP-seq peaks and the number of interactions both correlate with expression level). The enrichment calculation used by us, as well as the previous studies listed above, used the same (genome-wide) background model for calculating enrichment for each of the five different expression groups. Thus, the difference in ChIP-seq peak density/interaction count was not accounted for between genes falling in different expression groups. When these differences were accounted for, the increase in enrichment level was reduced.

Following the reviewer’s suggestion, we have now changed Figure 3 to focus less on this aspect of the data. As the reviewers pointed out, it is nevertheless the case that highly expressed genes contact enhancer regions with greater frequency than lowly expressed genes, and that this likely contributes to their increased expression levels. We have refocused Figure 3 to report that promoter interactions are enriched for histone marks that define enhancers and repressors, and that this enrichment correlates with expression level, as expected. Additionally, we re-calculated enrichment values by using the method reported in Schoenfelder et al., 2015, where the “expected” values were obtained by analyzing non-baited MboI fragments (i.e. not shuffling peaks). We also report that the correlation of histone ChIP-seq peaks with expression level is apparent even at large (300 kb) distances from promoters, an expansion of a phenomenon reported several years ago (Karlic et al., 2010).

We feel that the resulting Figure 3 is more intuitive and streamlined and thank the reviewers for their suggestions.

**Author response image 2. respfig2:** Correlation of expression with number of enhancer contacts. (**A**) Genes were grouped into 5 categories according to expression levels (q0=TPM 0, q1=TPM 0-3, q2=TPM 3-25, q3=TPM 25-150, q4=TPM>150) and the number of promoter-distal H3K27ac ChIP-seq peaks contacted by each promoter is displayed. The blue vertical bar indicates the median. (B,C) The median number of H3K27ac peaks contacted by promoters in each expression group is plotted against the expression group value for iPSC (**B**) and CM (**C**). Only promoter-distal interactions were considered. Spearman’s rho values are shown for the correlation estimate between expression and number of enhancers contacted. The same correlations were obtained when grouping genes by hard quantile cut-offs instead of TPM values, as in Schoenfelder et al. Genome Research 2015.

4) The paper would be further strengthened by providing examples t some of the predicted long-range interactions are functional. Are any of the predicted target genes affected in disease, e.g. reduced or increased in expression in individuals carrying the distal variant?

We have addressed this important point in several areas of our manuscript.

First, we show examples of genes that form long-range interactions to in vivovalidated heart enhancers (*GATA4* in Figure 1E and *NPPA* in Figure 3F) using the Vista enhancer database, the largest experimentally tested set of human enhancers in in vivoassays. We further assessed enrichment of the full repertoire of Vista enhancers in CM interactions and show here that heart enhancers are 3-fold enriched; importantly, the heart enhancer set is the most significantly enriched set of all 21 tissues characterized (Author response image 3).

Second, we show that predicted target genes of CVD SNPs, using CM interactions, are significantly enriched for causing cardiovascular phenotypes when the gene is knocked out in mice (Figure 5D), supporting that target genes have biological roles in cardiovascular disease.

Third, we show that human left ventricle eQTLs (from the latest GTEx analysis) are significantly enriched for looping to their eQTL-associated gene in CMs (Figure 5E); for this analysis, we used the stringent background of moving each promoter’s set of interactions to a new promoter, thereby retaining the overall clustering pattern of eQTLs (due to LD) and the often observed clustering of promoter interactions. Importantly, when we performed the same analysis using iPSC interactions, the observed proportion of eQTLs looping to their eQTL-associated gene was much less significant (Figure 5F).

Fourth, we report that the set of GWAS LD SNPs that falls within CM promoter interactions is enriched for eQTLs compared to the set of all GWAS LD SNPs (20% compared to 12%; Figure 5E). This result indicates that CM promoter interactions identify a subset of LD SNPs most likely to be functional, based on association with gene expression levels.

Fifth, we show in Figure 7D that the *SORT1* promoter contacts a myocardial infarction GWAS locus 120 kb away. This locus was experimentally dissected in Musunuru et al., 2010, where it was shown that the SNP rs12740374 introduces a binding site for the transcription factor C/EBP which resulted in increased enhancer activity of this region. Importantly, the allele associated with increased risk for myocardial infarction is the same allele that introduces the C/EBP binding site and is associated with increased expression of *SORT1*, consistent with the reporter assay results. The fact that the CM interaction map identifies multiple long-range interactions between the *SORT1* promoter and this distal myocardial infarction locus (including ~1 kb from SNP rs12740374, see Figure 7D) is an important level of evidence that this distal region is indeed a disease-associated regulatory element, and is supported by the functional characterization of the distal region in a previous study.

Although we did not experimentally validate the function of promoter interactions as part of this study, the multiple tiers of existing experimental results and orthogonal data sets strongly support the function of several specific interactions (e.g. interactions to Vista heart enhancers and the myocardial infarction locus), as well as large sets of interactions (e.g. interactions linking eQTLs to their eQTL-associated gene).

**Author response image 3. respfig3:** Enrichment of in vivovalidated enhancers (from the Vista Enhancer Browser) in CM promoter-distal interactions. Top, fold-enrichment of the observed number of enhancers compared to 1000 permutations of enhancer locations. Numbers above the error bars indicate the number of enhancer elements in each group. Bottom, corresponding Z-score for each enrichment. Heart enhancer data is highlighted in red.